# Stepping Forward on the Last Mile

**Chen Feng**
Qualcomm AI Research *
Qualcomm Canada ULC
chenf@qti.qualcomm.com

**Shaojie Zhuo**
Qualcomm AI Research *
Qualcomm Canada ULC
shaojiez@qti.qualcomm.com

**Xiaopeng Zhang**
Qualcomm AI Research *
Qualcomm Canada ULC
xiaopeng@qti.qualcomm.com

**Ramchalam Kinattinkara Ramakrishnan**
Qualcomm AI Research *
Qualcomm Canada ULC
rkinatti@qti.qualcomm.com

**Zhaocong Yuan**
Qualcomm AI Research *
Qualcomm Canada ULC
zhaocong@qti.qualcomm.com

**Andrew Zou Li**
University of Toronto
andrewzou.li@mail.utoronto.ca

## Abstract

Continuously adapting pre-trained models to local data on resource constrained edge devices is the *last mile* for model deployment. However, as models increase in size and depth, backpropagation requires a large amount of memory, which becomes prohibitive for edge devices. In addition, most existing low power neural processing engines (e.g., NPUs, DSPs, MCUs, etc.) are designed as fixed-point inference accelerators, without training capabilities. Forward gradients, solely based on directional derivatives computed from two forward calls, have been recently used for model training, with substantial savings in computation and memory. However, the performance of quantized training with fixed-point forward gradients remains unclear. In this paper, we investigate the feasibility of on-device training using fixed-point forward gradients, by conducting comprehensive experiments across a variety of deep learning benchmark tasks in both vision and audio domains. We propose a series of algorithm enhancements that further reduce the memory footprint, and the accuracy gap compared to backpropagation. An empirical study on how training with forward gradients navigates in the loss landscape is further explored. Our results demonstrate that on the last mile of model customization on edge devices, training with fixed-point forward gradients is a feasible and practical approach.

## 1   Introduction

On-device training allows pre-trained models to be continuously adapted to newly collected personal data after deployment. Moving model training from the cloud to local devices is essential for model customization and protecting users' privacy (Moon et al. [2024]). However, the constraint on power and memory makes training on edge devices extremely challenging (Dhar et al. [2019]). Traditional backpropagation involves a forward step, which computes activations given an input, and a backward step which computes the gradients. Intermediate activation values must be stored in memory prior to

---

*Qualcomm AI Research is an initiative of Qualcomm Technologies, Inc.

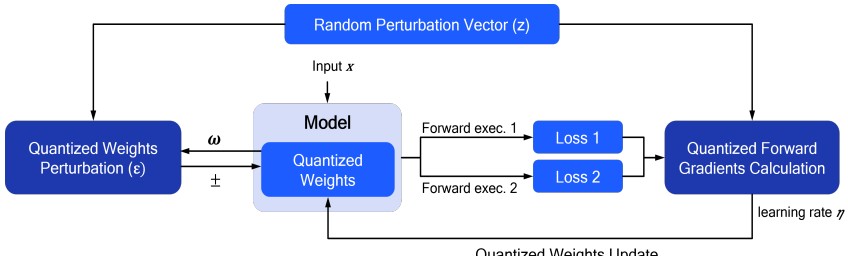

Figure 1: An overview of fixed-point forward gradient learning. The pipeline includes quantized weights perturbation, quantized forward gradient calculation through two forward calls with perturbed weights, and quantized weights update. Each process is explained in details in section 3.3.

the gradient of a certain layer is computed (Baldi and Sadowski [2016]). As models increase in size and depth, this process requires a prohibitive amount of memory for most existing edge devices.

To avoid large memory consumption, recent studies have re-examined the procedure of computing *Forward Gradients* as an alternative to standard backpropagation (Fournier et al. [2023]). As introduced by Baydin et al. [2022], a forward gradient, computed through a random, isotropic directional derivative, is an unbiased approximation of a weight gradient. Forward gradients can be further estimated solely with two forward calls of a neural network (Liu et al. [2020]), which saves computation and memory substantially. The work of MeZO (Malladi et al. [2023]) applies forward gradients on fine-tuning Large Lanuguage Models (LLMs), and shows a success on diverse downstream tasks, with the same memory footprint as inference.

Despite the aforementioned benefits, forward gradients may encounter the curse of dimensionality as the size of trainable parameters increases. Gradient approaximations from two forward calls may be noisy and with large variance (Ren et al. [2023]), resulting in less effective training of large networks. Moreover, most existing low power neural processing engines (e.g., NPUs, DSPs, MCUs, etc.) are designed as efficient fixed-point inference accelerators. The feasibility of utilizing fixed-point forward gradients for quantized training remains uncertain. Our goal is to gain deeper insights into whether training with fixed-point forward gradients can still result in competitive models while preserving the memory and computation benefits. To answer the question, we conduct comprehensive experiments across a variety of deep learning benchmark tasks in both vision and audio domains. A series of algorithm enhancements are proposed to further reduce the memory footprint, and accuracy gap compared to backpropagation. We believe our study to be of high interest in making model personalization happen locally on edge devices.

**Contributions.** **(a)** We formulate the computation of forward gradients in the quantized space. Weight perturbations and gradient calculations are all in fixed-point precision during model training or adaptation (see Figure 1 and Section 3). **(b)** We demonstrate the feasibility of on-device training with fixed-point forward gradients, through comprehensive experiments across a variety of deep learning benchmark tasks in both vision and audio domains. Although the method is model architecture agnostic, the experiments cover most typical model types (e.g., CNN, RNN, ViT-based) and parameter sizes (100K to 80M). **(c)** We propose a series of algorithm enhancements that further reduce the memory footprint and accuracy gap compared to backpropagation, leading to a practical solution for model adaptation on edge devices. **(d)** Finally, we visualize the neural loss landscape and trajectories of training with forward gradients, and show its dynamics and characteristics.

## 2 Related Work

### 2.1 Memory Efficient Training through Backpropagation

With an increasing number of applications using large neural networks on device, there is a demand of moving model training from the cloud to local devices. However, the key bottleneck for efficient on-device training is the limitation of memory resources. For example, training a simple Convolutional Recurrent model (CRNN, Keren and Schuller [2017]) with a parameter size of 250kB, requires 11.5MB ($46\times$) memory to store activations. Training memory is primarily attributed to activations

rather than parameters. Studies on algorithms to reduce resource consumption during training have been published, with a trade-off between memory usage and model accuracy. Parameter-efficient fine-tuning techniques such as LoRA (Hu et al. [2021]) and prefix tuning (Li and Liang [2021]) are proposed to train a model with reduced parameters. Dynamic sparse representation (Mostafa and Wang [2019]) is proposed to reduce memory requirements by making the weight and activation values sparse during training. Low precision training (Micikevicius et al. [2018]) reduces model sizes and computation requirements by adopting 16-bit float precision instead of 32-bit. The work of Lin et al. [2022] pushes conventional convolutional neural network training on devices with only 256kB by pruning the training graph during compilation time. These methods mainly focus on reducing the trainable parameters or activation sizes, thus reduce the peak memory required for training a neural network. However, due to the inherent nature of backpropagation, intermediate activations across all layers must be retained until loss is backpropagated and gradients are calculated. Therefore, as models increase in size and depth, parameter-efficient techniques do not fundamentally resolve the training memory problem.

## 2.2 Forward Gradients through Zeroth-order Optimization

Forward gradient has been recently brought to attention by Baydin et al. [2022] and Silver et al. [2022], which showed that gradients can be computed solely based on the directional derivatives using the forward mode of auto-differentiation only. The forward gradients can be estimated via two forward calls using zeroth-order optimization (Liu et al. [2020]) by incorporating random perturbations on weights, entirely eliminating the need for backpropagation in gradient descent. The work of Ren et al. [2023] shows that it is possible to substantially reduce the variance of the forward gradient estimation by applying perturbations to activations rather than weights. Considering the memory required for storage of intermediate activations, only weight-perturbed forward gradient estimator can be deployed on low resource constrained devices. While research by Belouze [2022] claimed shortcomings of forward gradients in high dimensions, the work of MeZO (Malladi et al. [2023]) proposes a contradictory perspective by showing the lower bounds of such zeroth-order optimization is conditioned on loss landscape instead of number of trainable parameters. MeZO further applies forward gradients on fine-tuning LLMs, and shows a success on diverse downstream tasks.

## 2.3 Quantized Training and Quantized Gradients

There is limited literature on gradient computation in the quantized space. Quantization-aware training (QAT Nagel et al. [2021]) has been widely used to simulate the potential quantization loss in the training stage. However, most existing low power neural processors (e.g., NPUs, DSPs, MCUs, etc.) are designed and optimized for fixed-point inference. Direct training in the quantized space will fundamentally bridge the gap between training and inference, thus being essential for model adaptation on edge devices. However, the work of Lin et al. [2022] observed that the quantization process distorts backward gradients, resulting in significantly lower accuracy in model training through backpropagation. Quantization-aware scaling (QAS) is proposed to address this problem. It remains uncertain whether training with quantized forward gradients through zeroth-order optimization can still lead to competitive models on device, while preserving the memory and computation benefits.

# 3 Quantized Forward Gradient Learning

Forward gradients utilize directional derivatives to bypass backpropagation, while retaining unbiased estimations of true gradients. In the following, we first review the technique of forward-mode autodifferentiation (AD Baydin et al. [2022]), alongside a practical implementation known as Simultaneous Perturbation Stochastic Approximation (SPSA) for zeroth-order gradient estimation (Spall [1992]). We then propose sign-m-SPSA, a variant of SPSA to alleviate the noisy component of forward gradients estimated by SPSA, which leads to stable performance in many use cases. Once the gradients are estimated, optimizers such as SGD, Adam etc. can be applied to update the weights. Finally, we formulate the Quantized Zeroth-order Forward Gradient (QZO-FF) estimator, mapping the processes of weights perturbation, gradients estimation and weights update in the fixed-point space. An overview of the QZO-FF algorithm is illustrated in Algorithm 1.

### 3.1 Forward Gradients

**Definition 1 (Forward Gradients).** Consider a machine learning function $f(w) : \mathbb{R}^n \to \mathbb{R}$, where $w \in \mathbb{R}^n$ is the trainable parameters that the gradients are evaluated. *Forward gradients* $g : \mathbb{R}^n \to \mathbb{R}^n$ is defined as:

$$g(w) = (\nabla f(w) \cdot z)z \qquad (1)$$

where $z \in \mathbb{R}^n$ is a perturbation vector taken as multivariate random variable $z \sim p(z)$ such that $z's$ scalar components $z_i$ are independent and have zero-mean and unit variance for all $i$. $\nabla f(w) \cdot z \in \mathbb{R}$, the Jacobian matrix-vector product, defines the directional derivative of $f$ at point $w$ in direction $z$.

### 3.2 Zeroth-order Optimization

In order to have runtime advantage over backpropagation, a classical zeroth-order estimator, SPSA can be used to estimate the forward gradients by evaluating $f$ in forward path $m$ times, where $m \ll n$.

**Definition 2 (SPSA).** Given a model $f$ with parameters $w \in \mathbb{R}^n$ and a loss function $\mathbb{L}(w)$, SPSA estimates the gradient as:

$$\hat{g}(w) = \frac{\mathbb{L}(w + \epsilon z) - \mathbb{L}(w - \epsilon z)}{2\epsilon} z \qquad (2)$$

where $z \sim \mathbb{N}(0, \mathbb{I}_n)$ is a weighted vector over all parameter dimensions, randomly sampled from normal distribution with zero-mean and standard deviation. The perturbation scale $\epsilon$ is a small constant value (e.g., $1e - 3$). For each sampled $z$, SPSA only requires two forward calls through the model, with positive and negative perturbed weights respectively, to estimate the gradients.

Gradient magnitude defined in (2) is determined by loss difference of two forward calls based on a random perturbation applied on weights, which easily becomes noisy. Inspired by many popular optimizers, such as sign-SGD and RMSProp (Bernstein et al. [2018]), updating weights through a sign-based method achieves good practical performance for many gradient compression use cases. In order to mitigate the noisy component of forward gradients estimated by SPSA, we propose sign-m-SPSA by only taking the direction of loss difference under a certain perturbation, while disregarding the magnitude component. The estimation can be improved by averaging $\hat{g}(w)$ over $m$ randomly sampled $z$ ($m \ll n$), with an increased number of training iterations.

**Definition 3 (Sign-m-SPSA).**

$$\hat{g}(w) = \frac{1}{m} \sum_{i=1}^{m} sign(\mathbb{L}(w + \epsilon z_i) - \mathbb{L}(w - \epsilon z_i))z_i \qquad (3)$$

The intuition behind sign-m-SPSA is that during the training, the estimator samples a random perturbation direction $z_i, i \in \{1, .., m\}$, and tests how it aligns with the true gradient by examining the loss change, and then multiplies the alignment direction with the perturbation direction. Weights will be updated along the sampled direction that leads to a decrease in loss. This design is also quantization-friendly, constraining the range of gradient values to be the same as perturbation for static quantization. Our later experiments show that 8-bit quantization of perturbation and forward gradient is sufficient for preserving the model accuracy across many use cases.

**Definition 4 (Sign-m-SPSA-SGD).** With $\hat{g}(w)$ as the forward gradients estimated through sign-m-SPSA, similar to backpropagation, an optimizer such as SGD with learning rate $\eta$ can be used to update model parameters:

$$w_{t+1} = w_t - \eta \hat{g}(w) \qquad (4)$$

### 3.3 Quantized Weights Perturbation and Forward Gradients

Sign-m-SPSA in (3) estimates forward gradients through a minimum of two forward calls, with positive and negative perturbed weights in float precision, respectively. For low power devices with fixed-point computation engines, model weights are quantized in low bit precision. Therefore, the random perturbation needs to be quantized prior to apply on weights.

For a given model, consider $w$ as the floating point weights of a certain layer. Assume model is per-tensor quantized with symmetric quantization in $b$-bit, the quantized weights can be represented by:

$$w_q = \lfloor \frac{w}{\Delta_w} \rceil \qquad (5)$$

where $\Delta_w$, denoted as the quantization scaling factor, is calculated by $\Delta_w = w_{max}/(2^{b-1} - 1)$, where $w_{max}$ is the maximum absolute value in $w$ found by a quantization method (i.e., TF, MSE, AdaRound, etc., Nagel et al. [2021]). $\lfloor \cdot \rceil$ represents for the rounding operation.

**Quantized Perturbation.** With the given quantization method in 5, the quantized weights perturbation can be defined and calculated as:

$$
\begin{aligned}
w \pm \epsilon z &= w \cdot 1.0 \pm \epsilon z \\
&\approx \Delta_w w_q \cdot \Delta_z \mathbf{1}_q \pm \Delta_w \epsilon_q \cdot \Delta_z z_q \\
&= \Delta_w \Delta_z (w_q \cdot \mathbf{1}_q \pm \epsilon_q \cdot z_q) \overset{re-quant}{\Longrightarrow} \Delta_w \cdot w_{q\pm}
\end{aligned}
\tag{6}
$$

Since weights $w$ and perturbation $z$ have different quantization scaling factors and possibly different bit-width used, we quantize 1.0 with the scaling factor of $z$, and quantize $\epsilon$ with the scaling factor of $w$, prior to direct adding the quantized values in accumulator. $\mathbf{1}_q = \lfloor \frac{1.0}{\Delta_z} \rceil$, represents for the quantized value of floating point 1.0 with $\Delta_z$ as its scaling factor. Similarly, $\epsilon_q = \lfloor \frac{\epsilon}{\Delta_w} \rceil$, represents for the quantized value of $\epsilon$ with $\Delta_w$ as its scaling factor.

The random perturbation vector $z$ is sampled from normal distribution with zero-mean and standard deviation $\mathbb{N}(0, \mathbb{I}_n)$, we can use static quantization with a pre-determined $z_{max}$ to pre-calculate $\Delta_z$. For example, in the case of $z_{max} = 3.5$, with 8-bit symmetric quantization, $\Delta_z = 0.0276$, and $\mathbf{1}_q = 36$. Similarly, $\epsilon_q$ can be pre-calculated, if a pre-trained model with $w_{max}$ is given. It is noted that $\epsilon$ is a very small value (e.g., $1e - 3$). Therefore, we require 16-bit to be used for weight quantization, such that $\epsilon$ can be properly represented by the minimum representation power of $\Delta_w$ without clipping loss, and small perturbation can be reflected on the weights change in the quantized space.

In (6), two values with 16-bit and 8-bit are multiplied, and then fed to a quantized add/subtract operation. In hardware, a 32-bit accumulator is used to hold the result. The result is then re-quantized to 16-bit by a multiply and a shift operation through a post-processing block (Appendix A), using the original weight scaling factor $\Delta_w$. The quantized perturbed weights are denoted as $(\Delta_w, w_{q+})$ and $(\Delta_w, w_{q-})$. The above formulation is derived under per-tensor quantization, however, per-channel quantization can be similarly derived with finer granularity.

**Quantized Forward Gradients.** Based on the quantization method in (5), quantized forward gradients, estimated from sign-m-SPSA, can be calculated as:

$$
\begin{aligned}
\hat{g}_f &= \frac{1}{m} \sum_{i=1}^{m} sign(\mathbb{L}(w + \epsilon z_i) - \mathbb{L}(w - \epsilon z_i)) z_i \\
&\approx \frac{1}{m} \sum_{i=1}^{m} sign(\mathbb{L}(w_{q+}) - \mathbb{L}(w_{q-})) \Delta_z z_q \\
&= \Delta_z g_q
\end{aligned}
\tag{7}
$$

where $g_q$ represents for the quantized gradients, and it is using the same quantization scaling factor and bit-width as perturbation vector $z$.

**Quantized Weights Update.** We can further quantize the learning rate $\eta$ to a quantized value of 1, using quantization scaling factor of $\Delta_\eta = \eta$. Finally, quantized weights update can be computed by:

$$
\begin{aligned}
w_{t+1} &= w_t - \eta \hat{g}_f \\
&\approx \Delta_w w_q - \Delta_\eta 1 \Delta_z g_q \\
&\approx \Delta_w w_q - \Delta_w \lfloor \frac{\Delta_\eta \Delta_z}{\Delta_w} g_q \rceil \\
&= \Delta_w (w_q - \bar{w}_q)
\end{aligned}
\tag{8}
$$

where $\bar{w}_q$ represents for the change of weights in the quantized space, with $\Delta_w$ as the re-quantized scaling factor (Appendix A). $\Delta_\eta$ can be pre-calculated. In our experiments, we find that it is important to keep weights in 16-bit, while the perturbation $z$ and gradient $g$ can be in 8-bit representations.

### 3.4 QZO-FF enhancement

**Momentum Guided Sampling.** Besides naive SGD, quantized forward gradient learning can also be combined with other optimizers such as Adam or SGD with momentum, with slight overhead to

---

**Algorithm 1** QZO-FF: Quantized Zero-order Forward Gradient Learning(quantized, fp16)

---
**Require:** quantized model parameters $w_q \in \mathbb{I}^n$, loss $\mathbb{L} : \mathbb{I}^n \rightarrow \mathbb{R}$, perturbation scale $\epsilon$, training steps $T$, batch size $B$, learning rate schedule $\{\eta_t\}$

1:   • Given a pre-defined $z_{max}$ of perturbation $z$, calculate $\Delta_z = z_{max}/(2^{b-1} - 1)$ with $b$-bit.
      • Quantize 1.0 to $\mathbf{1}_q$ with $\Delta_z$.
      • Get the quantization scaling factor, $\Delta_{w^i}$, of quantized weights of each layer.
2: **for** t = 1, ..., T **do**
3:     **for** m=1, ..., M **do**
4:         Sample random seed $s$, and batch $B$
5:         Generate perturbation vector $z \sim \mathbb{N}(0, \mathbb{I}_n)$, and quantize the values to $(\Delta_z, z_q)$, $z_q \in \mathbb{I}^n$
6:         $w_{q+} \leftarrow PerturbParameters(w_q, z_q, \epsilon_q)$          ▷ Perturb in positive direction
7:         $l_+ \leftarrow \mathbb{L}(w_{q+}; B)$
8:         $w_- \leftarrow PerturbParameters(w_q, z_q, -2\epsilon_q)$      ▷ Perturb in negative direction
9:         $l_- \leftarrow \mathbb{L}(w_{q-}; B)$
10:        $g_q^a += sign(l_+ - l_-) \cdot z_q$          ▷ Quantized gradient accumulation
11:        $w_q \leftarrow PerturbParameters(w_q, z_q, \epsilon_q)$      ▷ Reset weights to original position
12:     **end for**
13:     $g_q = g_q^a/M$                 ▷ Quantized gradient averaging
14:     **for** $w_q^i \in w_q$ **do**            ▷ Update weights of each layer
15:         $\bar{w}_q^i = \lfloor \frac{\Delta_\eta \Delta_z}{\Delta_{w^i}} g_q \rceil$   ▷ Re-quantization (see Append.A for fixed-point approximation)
16:         $w_q^i \leftarrow w_q^i - \bar{w}_q^i$
17:     **end for**
18: **end for**
19:
20: **Subroutine:** $PerturbParameters(w_q, z_q, \epsilon_q)$
21: **for** $w_q^i \in w_q$ **do**
22:     $w_q^i \leftarrow \lfloor \Delta_z(w_q^i \cdot \mathbf{1}_q + \epsilon_q \cdot z_q) \rceil$, where $\epsilon_q = \lfloor \epsilon/\Delta_{w^i} \rceil$       ▷ per-tensor $\Delta_{w^i}$
23: **end for**

---

store the gradient history. Similarly, by allocating additional memory to store the perturbation history, momentum can be used to guide the sampling process. Instead of sampling solely from a zero-centered Gaussian distribution, perturbations are computed from a combination of a momentum-centered and a zero-centered Gaussian distribution. Mathematically, $z_1 \sim \mathbb{N}(0, \mathbb{I}_n * \sqrt{\alpha})$, $z_2 \sim \mathbb{N}(z_t, \mathbb{I}_n * \sqrt{1 - \alpha})$, and $z_{t+1} = \beta * z_1 + (1 - \beta) * z_2$. Here, $\beta$ is a smoothing parameter; $\alpha$ and $\beta$ can be adaptively adjusted during training. For example, during the initial training stage, random perturbations are applied with $\beta = 1$. As training progresses, a history of the momentum $z_t$ is incorporated to guide the new sampling process.

**Sharpness-aware Perturbation.** Motivated by the connection between sharpness of the loss landscape and model generalization, we can perturb parameter values from its neighborhood location. This is done by performing an additional step of directional gradient ascent through parameter perturbation and loss evaluation, prior to QZO-FF, as illustrated in Figure 2. This process helps to prevent the model from converging to a sharp minimum.

**Sparse Update.** To further reduce memory consumption, the forward gradient learning can be combined with a sparsity algorithm such that only a subset of the weights are selected from the network for updating. Examples of sparsity algorithm may include pruning by top-$k$ magnitude, randomized pruning, pruning values beyond a specified threshold, to determine the importance of the weights. Our experiments show that incorporating sparsity with forward gradient learning allows for a $90\%$ reduction in the size of trainable parameters, with only minor decrease in accuracy, as well as slight improvement in convergence speed.

**Kernel-wise Normalization.** In (3), forward gradients are estimated through sign-m-SPSA. In addition, we can also apply a kernel-wise normalization to scale the gradient adaptively. $z$ is normalized by the norm of $w$ in each layer.

$$\hat{g}(w^i) = sign(\mathbb{L}(w + \epsilon z) - \mathbb{L}(w - \epsilon z))z^i/\|z^i\| \cdot \|w^i\| \tag{9}$$

# 4 Experiments

## 4.1 Few-shot learning

We first apply forward gradient learning in the setting of few-shot learning, targeting to adapt a base-learner to a new task for which only a few labeled samples are available. Experiments across a variety of challenging few-shot learning benchmarks in both vision and audio domains are explored. Models are trained for each dataset individually and then evaluated with the corresponding test split.

To address whether forward gradient learning (FF) could match the performance of backpropagation (BP), we explore classification tasks on training models with full fine-tuning (FT) and linear probing (LP), utilizing float16 (fp16) precision. Training accuracy with quantized FF (16-bit weights and 8-bit activations, 16w8a) is also evaluated and compared with that of fp16 precision. Details and analysis on memory usage during training are reported in Appendix B - E.

Table 1: Vision datasets used for few-shot learning

| Name | Setting | No. Classes (train/val/test) | No. Samples | Resolution |
|---|---|---|---|---|
| CUB | Bird Species | 200 (140/30/30) | 11,788 | $84 \times 84$ |
| Omniglot | Handwritten characters | 1623 (1000/200/423) | 32,460 | $28 \times 28$ |
| Cifar100_fs | Color | 100 (64/16/20) | 60,000 | $32 \times 32$ |
| miniImageNet | Natural images | 100 (64/16/20) | 60,000 | $84 \times 84$ |
| tieredImageNet | Natural images | 608 (351/97/160) | 779,165 | $84 \times 84$ |

Table 2: Vision tasks: few-shot learning accuracy (%) with Forward (FF) and Backward (BP) gradients. The averaged accuracy over 100 testing tasks is reported. FT: full fine-tuning; LP: linear probing; Quant: 16w8a with symmetric quantization. FF outperforms zero-shot across the board, and achieves comparable performance (accuracy within 5%) to BP on 26 out of 30 tasks.

| Backbone | Training | CUB | Omniglot | Cifar100_fs | miniImageNet | tieredImageNet |
|---|---|---|---|---|---|---|
| | Zero-shot | 68.46 | 92.00 | 60.44 | 84.44 | 80.92 |
| Resnet12 | BP, FT | **85.32** | **99.62** | **82.32** | 87.34 | **82.54** |
| | BP, LP | 84.14 | 98.64 | 72.42 | **87.46** | 81.96 |
| | FF, FT | 80.58 (-4.74) | 97.44 (-2.18) | 71.24 (-11.08) | 87.36 (+0.02) | 82.12 (-0.42) |
| | FF, LP | 79.02 (-5.12) | 96.62 (-2.02) | 70.30 (-2.12) | 87.30 (-0.16) | 82.22 (+0.26) |
| | FF, LP, Quant | 77.42 | 96.08 | 68.54 | 87.00 | 81.64 |
| | Zero-shot | 59.96 | 86.68 | 74.60 | 82.58 | 80.44 |
| Resnet18 | BP, FT | **79.28** | **98.54** | **86.34** | 86.96 | **86.78** |
| | BP, LP | 78.92 | 96.48 | 84.88 | 87.42 | 84.68 |
| | FF, FT | 76.34 (-5.64) | 94.70 (-3.84) | 82.20 (-4.14) | **87.66** (+0.70) | 85.88 (-0.90) |
| | FF, LP | 73.64 (-5.28) | 95.56 (-0.92) | 82.32 (-2.56) | 87.14 (+0.32) | 83.02 (-1.66) |
| | FF, LP, Quant | 70.54 | 95.86 | 74.92 | 85.74 | 81.00 |
| | Zero-shot | 90.60 | 90.96 | 82.28 | 98.78 | 94.30 |
| ViT tiny | BP, FT | 93.08 | **99.88** | **90.88** | 98.46 | **96.04** |
| | BP, LP | 93.90 | 95.78 | 84.42 | 98.40 | 95.32 |
| | FF, FT | **93.58** (+0.50) | 96.96 (-2.92) | 88.66 (-2.22) | **99.08** (+0.62) | 95.50 (-0.54) |
| | FF, LP | 92.26 (-1.64) | 95.00 (-0.78) | 84.48 (+0.06) | 99.02 (+0.62) | 95.18 (-0.14) |
| | FF, LP, Quant | 92.24 | 95.04 | 84.40 | 99.00 | 95.18 |

**Vision Benchmark.** Image classification models are compared across commonly used 5 few-shot learning benchmark datasets (Table 1). Training methods are evaluated on 3 network backbones (modified Resnet12 Ye et al. [2020], Resnet18 He et al. [2015] and ViT tiny Dosovitskiy et al. [2020]), with ProtoNets Snell et al. [2017] as few-shot classifier.

Table 2 demonstrates the classification accuracy on vision benchmarks. We first show that FF significantly improves over zero-shot performance across model types and tasks. Given that FF solely utilizes directional derivatives for gradient estimation, it is expected that BP generally outperforms FF in most tasks. The accuracy gap between BP and FF can vary based on factors such as backbone architecture, dataset and task difficulty. The largest accuracy degradation is observed when training Resnet12 on Cifar-100 dataset with an input resolution of $32 \times 32$. However, using a stronger backbone such as ViT, can help bridge this accuracy gap. This indicates that while FF may show some degradation with smaller architectures and low-resolution inputs, performance improvements can be achieved with more advanced models. Overall, FF achieves comparable performance (accuracy within 5%) to BP in 26 out of 30 comparable experiments. A minimal accuracy drop is observed in quantized FF training, when a strong backbone such as ViT tiny is used. These promising results

Table 3: Audio datasets used for few-shot learning. The ESC-50 dataset includes a labeled collection of 2000 environmental audio recordings, and FSDKaggle2018 is an audio dataset containing 11,073 audio files annotated with 41 labels of the AudioSet Ontology. Both datasets are used for benchmarking methods of environmental sound classification.

| Name | Setting | No. Classes (train/val/test) | No. Samples | Sample Length |
|------|---------|------------------------------|-------------|---------------|
| ESC-50 | Environmental | 50 (35/5/10) | 2,000 | 5s |
| FSDKaggle18 | Mixed | 41 (29/5/7) | 11,073 | 0.3s - 30s |

indicate that FF can perform comparably to BP with only a slight degradation in accuracy, while significantly reducing the memory cost (see analysis in Appendix B.1). With the same memory footprint as inference, model training with FF is feasible on low memory devices where BP cannot be afforded.

**Audio Benchmark.** Two audio benchmark datasets (ESC-50 and FSDKaggle18) are selected (Table 3) for sound classification use cases using few-shot learning. Similar to vision, training methods are evaluated on 2 representative architectures CRNN (Heggan et al. [2022]) and Audio Spectrogram Transformer (AST Gong et al. [2021]), with SimpleShot (Wang et al. [2019]) and ProtoNets (Snell et al. [2017]) as few-shot classifiers.

Table 4: Audio tasks: few-shot learning accuracy (%) with Forward (FF) and Backward (BP) gradients. FF achieves comparable (accuracy within $5\%$) or better performance to BP on 11 out of 16 tasks.

| Backbone | Training | ESC-50 | | FSDKaggle18 | |
|----------|----------|------------|----------|------------|----------|
| | | SimpleShot | ProtoNet | SimpleShot | ProtoNet |
| CRNN | BP, FT | 66.34 | **73.82** | **38.89** | 33.11 |
| | BP, LP | **72.11** | 71.30 | 36.88 | 32.67 |
| | FF, FT | 67.20 (+0.86) | 64.30 (-11.39) | 36.04 (-2.85) | 35.52 (+2.41) |
| | FF, LP | 67.38 (-4.73) | 61.62 (-9.68) | 37.53 (+0.65) | 34.67 (+2.00) |
| | FF, LP, Quant | 67.05 | 63.43 | 36.90 | **35.55** |
| AST | BP, FT | 68.04 | **75.85** | 38.12 | **46.12** |
| | BP, LP | 75.98 | 70.16 | 42.86 | 42.64 |
| | FF, FT | **79.70** (+11.66) | 66.98 (-8.87) | **42.92** (+4.80) | 40.50 (-5.62) |
| | FF, LP | 76.07 (+0.09) | 63.96 (-6.20) | 42.72 (-0.14) | 38.18 (-4.46) |
| | FF, LP, Quant | 76.13 | 61.86 | 42.90 | 38.10 |

Table 4 reports classification accuracy on audio benchmarks. Compared to vision tasks, the accuracy gap is larger, ranging from $-11.39\%$ to $+11.66\%$. This may be due to the extremely challenging training setting of 5-way 1-shot, where only 1 example of each class is seen in each task. Additionally, we found that the pretrained model from AudioSet (AST) does not produce a good zero-shot performance across all tasks. This indicates that a good initial baseline is critical for model adaptation. Overall, FF achieves comparable (accuracy within $5\%$) or better performance to BP on 11 out of 16 tasks. Training with quantized FF (16w8a) maintains similar accuracy level as fp16. From memory analysis in Appendix B.2, training an AST model with quantized forward gradients combined with sparse update, requires only 0.19MB scratch memory, which fits into most existing edge devices.

## 4.2 Cross-domain Adaptation

We further conduct experiments on model adaptation to cross-domain datasets, in which a models is fine-tuned on tasks with data distribution significantly different from those of the pre-trained model. For ablation studies on various impacts on the training accuracy, we take ViT tiny (5.5M parameters) as backbone for feature extractor, and apply a randomly initialized linear layer as the decoder for binary classifier. The model is pretrained on ImageNet-1k through DeiT (Touvron et al. [2021]), and adapted for Visual Wake Word (VWW) task (Chowdhery et al. [2019]) through linear probing (LP), where only the decoder layer is fine-tuned, and visual-prompt tuning with deep prompts (D-VPT, Jia et al. [2022]), where prompts in each Encoder layer are also fine-tuned. Testing accuracy is reported in Figure 2, and detailed training hyper-parameters are listed in Appendix C.

**Effectiveness of Quantized FF.** With LP, quantized forward gradient learning is capable of training the model to an accuracy of $87.30\%$ from $48.50\%$, with an accuracy gap of $0.63\%$ compared to BP in fp16.

**Gradient averaging in FF.** A larger $m$, used to average forward gradients, helps to smooth the noisy estimation and increases the model accuracy. With D-VPT training in fp16, simply increasing $m$ to 3

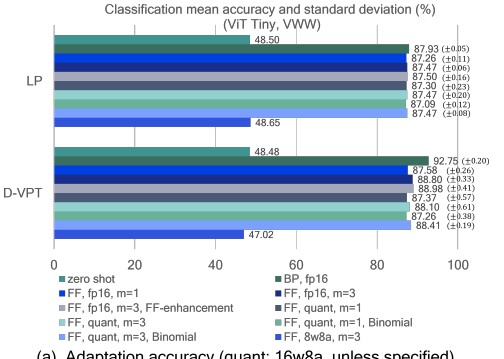

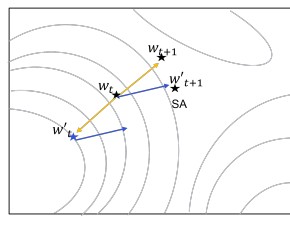

(b) Schematic of sharpness-aware update. weights perturbation at neighborhood position.

(a). Adaptation accuracy (quant: 16w8a, unless specified).

Figure 2: Ablation studies on cross-domain adaptation. The accuracy numbers (with standard deviation) are averaged over 5 runs.

boosts the accuracy by $1.22\%$. However, there is a trade-off between model accuracy and training efficiency.

**Quantization bit-width.** Experiments show that 8-bit weights quantization (8w8a) does not lead to model convergence. Therefore, 16-bit weights quantization is necessary to capture the small perturbation, while the perturbation $z$ and gradients can use 8-bit.

**Perturbation sampling.** The random perturbation $z$ in Equation (2) is sampled from a normal distribution with zero-mean and standard deviation $\mathbb{N}(0, \mathbb{I}_n)$. Other distibutions, such as Binomial distribution, also works well for forward gradient learning.

**QZO-FF enhancement.** FF can be extended with sharpness-aware scheme, where a perturbation is performed at a neighborhood location through an extra step of gradient ascent. Together with kernel-wise normalization, this technique results in the closest performance to BP in both training methods. Although obtaining the norm of weights involves a trade-off between computation and accuracy, efficient implementations using *gemm* and *sqrt* operations can minimize the overhead on hardware.

**Loss landscape.** It is believed that the convergence and generalization property of perturbation-based learning, such as forward gradient learning, depends on the loss landscape instead of number of parameters. Visualization of loss landscape has the potential to help us answer several important questions about how a neural network is trained, and why do the resulting minima generalize under certain training approach. Utilizing the tool provided in Li et al. [2018], we show the 2D contours of loss landscape of ViT tiny network under the task of cross-domain adaptation, together with the loss trajectory during training, providing an empirical characterization of neural loss functions, and exploring how training with forward gradients navigates in the loss landscape (See Appendix E).

### 4.3 In-domain OOD Adaptation

On-device model adaptation often involves fine-tuning on data that is out-of-distribution (OOD). To evaluate the performance of FF, we pretrain a ViT tiny backbone on Cifar10, and fine-tune the decoder on Cifar10-C (Hendrycks and Dietterich [2019]), where 15 types of corruptions, such as Gaussian noise or pixelation, of varying severity are applied. We take the lowest (easy), middle (medium), and highest (hard) corruption severity from the dataset as separate benchmarks for fine-tuning. Fine-tuning techniques include LP with 1 linear decoder layer, LP with 3 linear decoder layers, and D-VPT (Jia et al. [2022]). Additionally, we explore the impact of sparsity by pruning $90\%$ of the trainable parameters using a zero-order method (Chen et al. [2024]). Table 5 shows a comparison of accuracy on the test set between BP, FF, quantized FF and Sparsed FF, alongside different fine-tuning methods. Detailed training hyper-parameters are listed in Appendix D.

As the number of trainable parameters increases, forward gradient learning improves the model accuracy on OOD dataset. Even with a sparsity level of $90\%$, FF can still achieve comparable accuracy levels to those of BP. The largest accuracy disparity between the two is $6.98\%$, observed on the Cifar10-C (hard) category using the LP method for 3 decoder layers. As corruption intensifies,

Table 5: Accuracy (%) of model adaptation to in-domain OOD dataset with Forward (FF) and Backward (BP) gradients. 1 LN: 1 linear layer of decoder; 3 LN: 3 linear layer of decoder. Quant: 16w8a, Sparse: 90% weights pruned. The accuracy numbers (with standard deviation) are averaged over 5 runs.

| Backbone | Training | Cifar10-C (easy) | Cifar10-C (median) | Cifar10-C (hard) |
|----------|----------|------------------|--------------------|--------------------|
| LP 1 LN | Zero-shot | 82.48 | 74.59 | 62.40 |
| | BP | 83.75 ($\pm$ 0.67) | 77.88 ($\pm$ 0.85) | 70.03 ($\pm$ 1.20) |
| | FF | 83.37 ($\pm$ 0.60) | 77.04 ($\pm$ 0.66) | 68.65 ($\pm$ 0.70) |
| | FF, Sparse | 83.34 ($\pm$ 0.59) | 77.11 ($\pm$ 0.68) | 68.63 ($\pm$ 0.95) |
| | FF, Quant | 83.23 ($\pm$ 0.57) | 76.73 ($\pm$ 0.75) | 68.28 ($\pm$ 0.87) |
| LP 3 LN | Zero-shot | 85.83 | 77.77 | 62.25 |
| | BP | 86.99 ($\pm$ 0.41) | 81.57 ($\pm$ 0.78) | 74.76 ($\pm$ 0.90) |
| | FF | 86.11 ($\pm$ 0.59) | 79.17 ($\pm$ 0.70) | 67.78 ($\pm$ 0.72) |
| | FF, Sparse | 86.10 ($\pm$ 0.58) | 79.24 ($\pm$ 0.63) | 68.06 ($\pm$ 1.11) |
| | FF, Quant | 85.77 ($\pm$ 0.55) | 78.67 ($\pm$ 0.63) | 67.25 ($\pm$ 0.42) |
| D-VPT | Zero-shot | 89.52 | 82.24 | 68.95 |
| | BP | 91.66 ($\pm$ 0.50) | 88.90 ($\pm$ 0.46) | 84.54 ($\pm$ 0.42) |
| | FF | 90.58 ($\pm$ 0.53) | 86.21 ($\pm$ 0.49) | 78.38 ($\pm$ 0.80) |
| | FF, Sparse | 90.56 ($\pm$ 0.48) | 86.18 ($\pm$ 0.51) | 78.24 ($\pm$ 0.81) |
| | FF, Quant | 90.41 ($\pm$ 0.49) | 85.77 ($\pm$ 0.43) | 77.45 ($\pm$ 0.64) |

the loss surface becomes less smooth, potentially causing FF to be impacted more from the noisy gradient estimation.

## 5  Conclusion

Continuously updating pre-trained models to local data on the edge is the last mile for model adaptation and customization. To overcome the memory limitation of most existing low power devices, forward gradients are used for model adaptation. We have formulated the forward gradient learning in the quantized space, where weight perturbations and gradient calculations are all in fixed-point during model training. To investigate the feasibility of on-device training with fixed-point forward gradients, we have extensively conducted experiments across a variety of deep learning benchmark tasks in both vision and audio domains. Model adaptation to cross-domain dataset and in-domain OOD datasets are further evaluated and analyzed.We further explore 2D contours of loss landscape, together with loss trajectory during training, providing an empirical explanation on how the model is trained. We have shown that quantized forward gradient learning with 16w8a can effectively adapt most typical model architectures (e.g., Resnet, ViT-tiny, CRNN, AST) and scales. With minimum accuracy reduction, fixed-point forward gradients allows model adaptation using the same memory footprint and operation support as inference, as opposed to backpropagation. Therefore, it has the potential to enable model fine-tuning on existing edge devices with limited memory and backpropagation support, without requiring additional hardware adaptation.

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

# A Fixed-point re-quantization

The process of quantized perturbation (Equation 6) and gradient calculation (Equation 8) involves a re-quantization process. In fixed-point engines, this is approximated by a multiply and a shift operation through a post-processing block.

$$w_q = \Delta_z(w_q \cdot \mathbf{1}_q + \epsilon_q \cdot z_q)$$
$$= (w_q \cdot \mathbf{1}_q + \epsilon_q \cdot z_q) \cdot m \gg k \tag{10}$$

where $m$ and $k$ are integer numbers, and $\frac{m}{2^k} \approx \Delta_z$.

# B Few-shot learning experiments

In our experiments, the number of forward-forward calls performed ($m$) for averaging gradients is 3 unless specified. All our experiments are running on single Nvidia Tesla V100 GPU. It is noted that our experiments do not aim to beat the benchmark state-of-the-art (SOTA) performance, but to compare the performance gap between forward and backward gradient learning across datasets and tasks. Due to the limited tuning performed, it is possible to obtain a specific result marginally better than those presented. However, this does not undermine the comparision investigated in this work.

## B.1 Vision Tasks

In vision benchmark, five common few-shot learning datasets are explored: CUB ([41]), Omniglot ([20]), Cifar100_fs ([6]), miniImageNet ([40]) and tieredImageNet ([31]). Each dataset is split into three parts based on different non-overlapping sets of classes, for model training, validation, and testing. All recognition tasks across datasets are using 5-way 5-shot setting.

Table 6: The hyper-parameters used in our few-shot learning experiments for vision tasks. For fair comparisons, FF and BP are using the same hyper-parameters. Model architectures of Resnet18, modified Resnet12 and ViT tiny are based on [14], [43], and [39]. Pre-trained models used for zero-shot evaluation can be found at [33], [34] and [38]. Different learning rate grids are explored, and the best accuracy is reported.

| Experiment | Hyper-parameters | Values |
|---|---|---|
| | n_way | 5 |
| | n_shot | 5 |
| | $\epsilon$ | 1e-3 |
| FF, BP | Epochs | 40 |
| | Optimizer | SGD |
| | Learning rate | {1e-3, 1e-4, 1e-5} |
| | Val/test tasks | 100/ 100 |

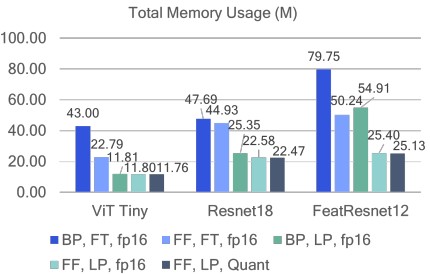
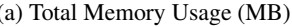

(a) Total Memory Usage (MB)

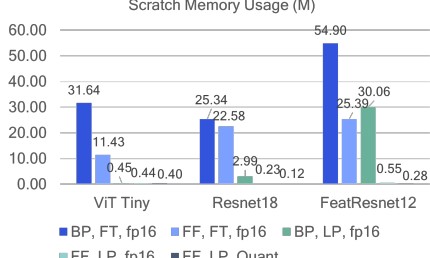

(b) Scratch Memory Usage (MB)

Figure 3: Comparison of Memory Usage during Training. BP: backpropagation, FF: forward gradient learning, fp16: 16-bit float point, Quant: 16w8a, FT: full fine-tuning, LP: linear probing.

Figure 3 shows the memory usage of BP and FF during the training. The total memory usage during training is composed of two parts, a scratch buffer used for input and output activation tensors for

gradient calculation and storage, and allocated memory for weights storage. Without storing the activation tensors, forward gradient learning has a significant reduction on the scratch memory usage. For example, in the case of full fine-tuning on ViT Tiny network, under the same precision of fp16, FF reduces the scratch memory from $31.64$MB to $11.43$MB ($2.8\times$). When sparse update and fixed-point training are enabled, only $0.40$MB of scratch memory is needed for model fine-tuning.

The extent of memory saving with FF depends on the number of layers being fine-tuned, and their positions within the network. When applied to methods such as full fine-tuning, LoRA ([17]) and other parameter-efficient fine-tuning approaches, FF shows significant memory reduction because it eliminates the need to store intermediate activations. In the case of LP, where only the last few layers are updated, the difference of memory usage between BP and FF will get smaller. As the number of trainable layers increases, FF benefits more in memory savings. These promising results indicate that FF can perform comparably to BP with only a slight degradation in accuracy, while significantly reducing the memory cost. With the same memory footprint as inference, model training with FF is feasible on low memory devices where BP cannot be afforded.

## B.2  Audio Tasks

In audio use cases, two few-shot audio classification benchmark datasets are selected: ESC-50 ([30]) and FSDKaggle18 ([11]). Prior to adaptation, publicly available pretrained models based on AudioSet are adopted ([1]). The averaged accuracy after 200 epochs over $10,000$ tasks drawn from the test set is reported.

Table 7: The hyper-parameters used in our few-shot learning experiments for audio tasks. Both datasets are using $5$-way $1$-shot setting. For fair comparisons, FF and BP are using the same hyper-parameters except that FF uses a smaller learning rate. Model architectures of CRNN and AST are based on [15] and [13]. Pre-trained models used for zero-shot evaluation can be found at [15] and [1]. Different learning rate grids are explored, and the best accuracy is reported.

| Experiment | Hyper-parameters | Values |
|---|---|---|
| | n_way | 5 |
| | n_shot | 1 |
| | $\epsilon$ | 1e-3 |
| FF, BP | Epochs | 200 |
| | Optimizer | SGD |
| | Learning rate | {1e-4, 1e-5} |
| | Val/test tasks | 100/ 10,000 |

Figure 4 compares the memory usage of BP and FF during the training. For a small model such as CRNN, there is at least $4\times$ reduction in total memory when full fine-tuning is used. In the case of AST architecture, model training with quantized forward gradient combined with sparse update only requires $0.19$MB scratch memory, which fits into most existing edge devices.

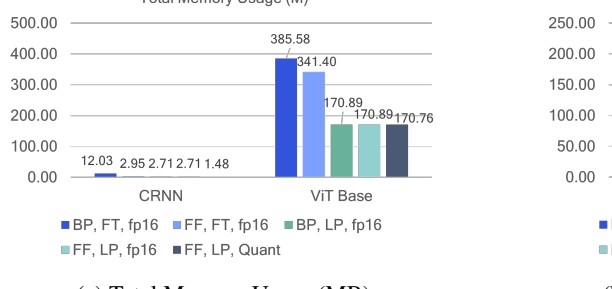

(a) Total Memory Usage (MB)        (b) Scratch Memory Usage (MB)

Figure 4: Comparison of Memory Usage during Training. BP: backpropagation, FF: forward gradient learning, fp16: 16-bit float point, Quant: 16w8a, FT: full fine-tuning, LP: linear probing.

# C   Cross-domain adaptation

Cross-domain adaptation is performed on VWW dataset. Table 8 lists all hyper-parameters used in training.

Table 8: The hyper-parameters used in our experiments for cross-domain adaptation. All hyper-parameters for FF and BP are the same except that FF uses a smaller learning rate. Model architectures of ViT tiny, and the associated pre-trained weights can be found at [39]. Different learning rate grids are explored, and the best accuracy is reported.

| Experiment | Hyper-parameters | Values |
|---|---|---|
| | $\epsilon$ | 1e-3 |
| | Epochs | 100 |
| | Warmup epochs | 20 |
| | Optimizer | Adamw, betas: [0.9,0.95] |
| FF, BP | Learning rate | {5e-3, 1e-3} |
| | Minimum learning rate | 1e-5 |
| | Scheduler | cosine decay |
| | Batch size | 256 |
| | Weight decay | 0 |

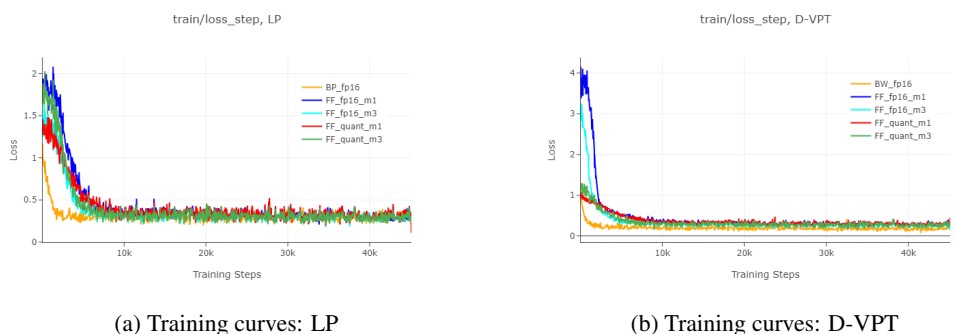

(a) Training curves: LP                      (b) Training curves: D-VPT

Figure 5: Training convergence curves. BP: backpropagation, FF: forward gradient learning, fp16: 16-bit float point, Quant: 16w8a, LP: linear probing, D-VPT: visual-prompt tuning with deep prompts.

Figure 5 shows the training curves of BP and FF under various settings. In general, FF requires a smaller learning rate, resulting more training iterations to converge than BP. However, for a single iteration, BP performs one forward pass and one backward pass, while FF needs two forward passes. The FLOPs of a backward pass are $\sim 2\times$ of that of a forward pass (e.g., for both Convolutional and Linear layers). Therefore, FF has a $1.5\times$ speedup in one iteration of the training. The total training time depends on the number of iterations required for model convergence and the time taken to complete each iteration.

# D   In-domain OOD adaptation

Cifar10-C provides 5 levels of corruption severity, from which we take the lowest (easy), middle (medium), and highest (hard) corruption severity as separate benchmarks for fine-tuning, randomly partitioning each section into a 90%-10% train-test split.

# E   Empirical Studies, Discussions and Limitations

The convergence and generalization property of perturbation-based learning, such as forward gradient learning, depends on the loss landscape instead of number of parameters. Visualization of loss landscape has the potential to help us answer several important questions about how a neural network is trained, and why do the resulting minima generalize under certain training approaches.

Table 9: The hyper-parameters used in our experiments for in-domain OOD adaptation. All hyper-parameters for FF and BP are the same, except that FF uses a smaller learning rate. Model architectures of ViT tiny, and the associated pre-trained weights can be found at [39]. Different learning rate grids are explored, and the best accuracy is reported.

| Experiment | Hyper-parameters | Values |
|---|---|---|
| FF, BP | $\epsilon$ | 1e-3 |
| | Epochs | 100 |
| | Warmup epochs | 0 |
| | Optimizer | Adamw, betas: [0.9,0.95] |
| | Learning rate | {1e-4, 5e-5, 1e-5} |
| | Minimum learning rate | 1e-5 |
| | Scheduler | cosine decay |
| | Batch size | 256 |
| | Weight decay | 0 |

Figure 6 compares the 2D contour of loss landscape and loss trajectory during training under BP and QZO-FF. Both forward and backward learning shows a locally smooth loss contour, and the trajectory follows the gradient descent direction, with forward gradient learning taking a more conservative step after each epoch, resulting in slower convergence. We also observed that a good initialization (e.g., pre-trained model) is critical for forward gradient learning. Therefore, the convergence may not be guranteed if a model is trained from scratch. However, it is still promising that quantized forward gradients to be used for model adaptation on low resource devices, in which a general pre-trained model has been deployed.

In our experiments, it is also observed that 8-bit quantization of weights does not lead to model convergence. This is because the small perturbation of $\epsilon$ is quantized using the scaling factor of weights ($\Delta_w$). It requires higher bits to be properly represented without clipping loss, thus the weights change can be reflected in the quantized space. In future, techniques for ultra low bit (i.e., 8-bit, 4-bit) forward gradient learning can be explored. In addition, experiments beyond classification and across multiple modalities can be conducted for further evaluations.

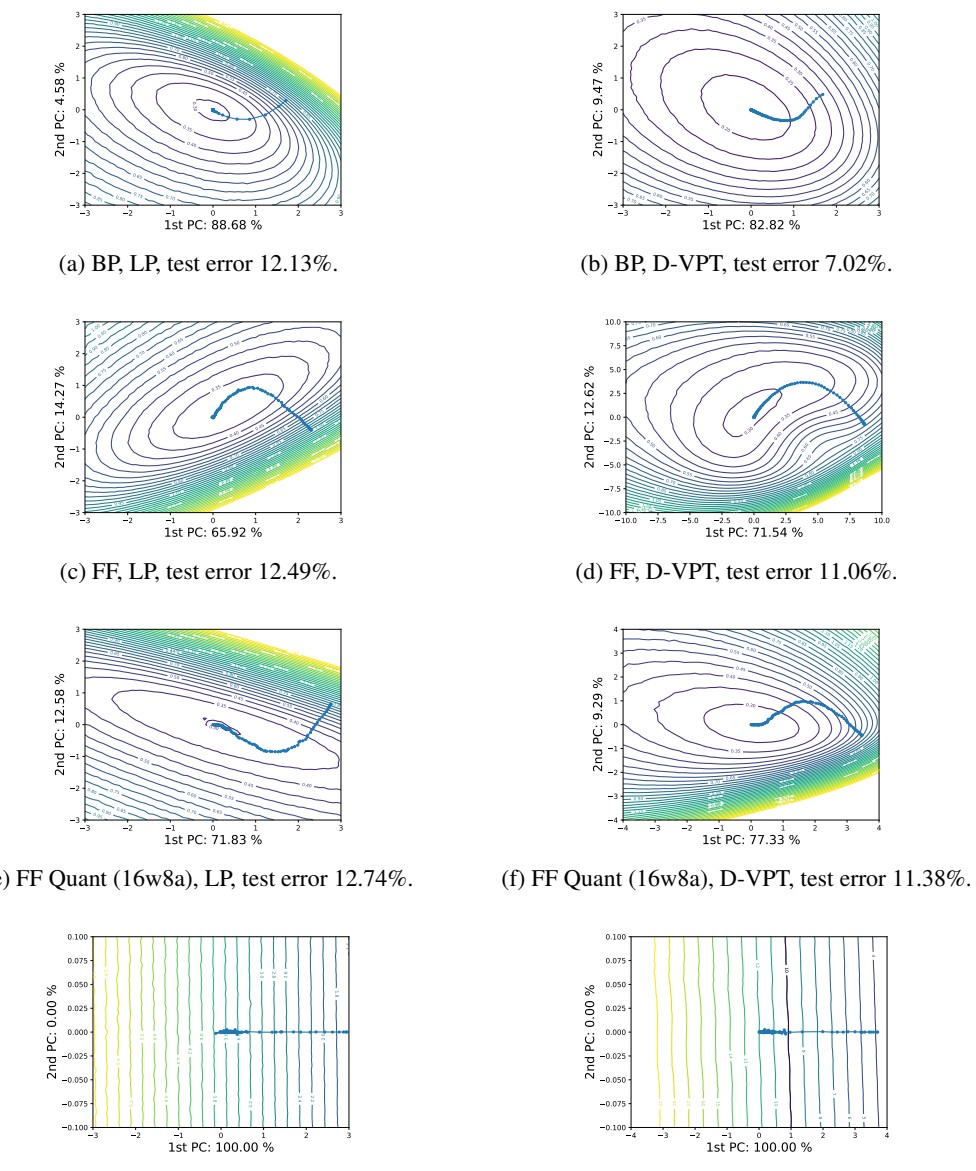

(a) BP, LP, test error 12.13%.

(b) BP, D-VPT, test error 7.02%.

(c) FF, LP, test error 12.49%.

(d) FF, D-VPT, test error 11.06%.

(e) FF Quant (16w8a), LP, test error 12.74%.

(f) FF Quant (16w8a), D-VPT, test error 11.38%.

(g) FF Quant (8w8a), LP, test error 51.35%, not converged.

(h) FF Quant (8w8a), D-VPT, test error 52.98%, not converged.

Figure 6: 2D visualization of loss landscape and loss trajectory during training. All hyper-parameters used in this experiment is listed in Appendix D. LP: linear probing, D-VPT: visual-prompt tuning with deep prompts. Both forward and backward learning shows a locally smooth 2D loss contour, and the trajectory follows the gradient descent direction, with FF taking a more conservative step after each epoch. It is observed that 8-bit quantization of weights does not lead to model convergence. Therefore, 16-bit weights quantization is necessary for QZO-FF.

