# OpenReview forum: "Stepping Forward on the Last Mile"
_NeurIPS.cc/2024/Conference — NeurIPS 2024 poster_

### Official Review · Reviewer_uc7y · 2024-07-04

**Soundness:** 3
**Presentation:** 3
**Contribution:** 3
**Rating:** 7
**Confidence:** 3

**Summary:**

**Context**. The focus of the present paper is on-device fine-tuning (gradient computation and weight update **starting from a pre-trained model**) under limited memory budget. One way to cut the memory cost of storing the computational graph for gradient computation by standard backprop is the Memory Efficient Zeroth Order (MeZO) optimizer [Malladi et al, 2023], whereby a directional gradient is computed via weight perturbation (a.k.a SPSA): computing the loss $L$ difference yielded by two forward passes with weights differing by $\epsilon u$ estimates $\nabla L \cdot u$. Since it is a purely forward procedure, it obviates the need to cache activations to execute a backward pass.

**Core contribution**. The present paper proposes a quantized version of MeZO where weight perturbation, gradient computation and weight update are carried out on quantized quantities. The proposed algorithm, coined QZO-FF (Alg. 1), is tested against a variety of fine-tuning tasks (few-shot learning, cross-domain and in-domain adaptation), modalities (image and audio data), and architectures (convolutional, attention-based, recurrent), with several variants being explored (with fp8 / fp32 activations) and benchmarked against standard backprop. The efficiency of QZO-FF, both in terms of resulting performance and memory usage, is demonstrated.

**Paper outline**. More precisely:
- Section 2 provides background knowledge on memory-efficient backprop (2.1), forward-mode differentiation (2.2) and quantized training (2.3).
- Section 3.1 and 3.2 formalizes further "forward gradients" (3.1)  and the SPSA / weight perturbation procedure to estimate them (3.2). A hardware-friendly extension of SPSA coined as "Sign-m-SPSA", which estimates $\text{sign}(\nabla L \cdot u) u$, , is introduced along with the resulting SGD update (3.2).
- Section 3.3 presents the core algorithmic contribution by combining SPSA / weight perturbation and weight quantization (Alg. 1). More precisely, weights and perturbation are statically, symmetrically quantized (e.g. their range are estimated and set once, before fine-tuning), with one scale for each ($\Delta_w$ and $\Delta_q$). Therefore: i) $\Delta_w$ and $\Delta_q$ are fixed, with weights and perturbations quantized with 16 and 8 bits respectively, ii) the integer part of the perturbed weights is accumulated in 32 bit, iii) the dequantized perturbed weights is quantized-dequantized back into 16 bits using the same $\Delta_w$ scale (Eq. 6). The Sign-m-SPSA gradient estimator is applied and quantized-dequantized using the perturbation scale ($\Delta_z$, Eq. 7) . Finally, the weight update itself is quantized, such that it happens in the quantized integer part and is rescaled by $\Delta_w$ (Eq. 8). Alg. 1 summarizes the procedure in the case where the number of perturbed directions at use is 1 ($m=1$).
- Section 3.4 presents several algorithmic "enhancements" of the QZO-FF algorithm to improve the optimization procedure itself or its memory footprint.
- Section 4 presents experimental results. First, few-shot learning is considered (4.1) on visual and audio data. Here, "FF" refers for short to "QZO-FF". A quantized version of FF, where 8 bits activations are used, is also tested. On vision, three architectures are tested (ResNet12, 18 and ViT tiny) on 5 different standard few-short learning datasets. Two scenarii are considered: full fine-tuning and linear probing. It is shown overall that FF always yields better performance than the zero-shot baseline and within 5%, accuracy-wise, to the BP baseline on 26/30 experiments, and that the ViT backbone yields the least degradation. On audio, a similar experiment is done with two architectures (CRNN, AST) on two audio datasets. On 11/16 experiments, FF accuracy is 5% off the BP baseline. Then, a cross-domain adaptation task (4.2) is considered, where the different algorithmic enhancements previously introduced (e.g. quantized FF, gradient averaging, "sharpness aware" scheme...) are tested. Most importantly, it is observed that quantizing weights to 8 bits jeopardize the FF algorithm. Finally, sector, 4.3 presents in-domain OOD adaptation using the same fine-tuning schemes (LP, D-VPT) with three levels of corruptions of the CIFAR-10 dataset as OOD datasets. In this setting, FF achieves comparable performance with BP.

**Strengths:**

- The problem tackled is highly relevant to on-device training, pragmatic and builds upon recent work [Malladi et al, 2023].
- The proposed algorithm is sound and well-explained.
- There are a lot of experimental settings, data modalities and architectures being explored.
- The proposed technique is effective in providing a learning signal, effectively training models and yielding relatively good performance compared to the BP baseline.

**Weaknesses:**

- It is unclear what is kept in full precision in the proposed procedure (see my questions below).
- On a related note, it is also unclear that the proposed algorithm enhancements don't offset the advantages of manipulating statically quantized quantities (see my questions below).
- The experiments aren't all sufficiently well explained, neither in the main nor in the appendix, which is frustrating because there is a lot of work done there and we fail to deeply understand the proposed setups. I would even say that there are almost too many different experimental setups. Under constrained time budget to write the paper, I would have prioritized a lesser number of better detailed experiments rather than a lot of them left unsufficiently explained.
- **There aren't any error bars in any table and figures**, although the authors ticked in their checklist that they reported error bars and provided appropriate information about the statistical significance of their experiments (L. 520). For lack of this, it is very hard to draw any clear conclusion in terms of comparison between the different algorithms at use, e.g. is there a statistically significant gap here, or are these two results within error bar? We don't know.
- I don't understand what the 2D plot of the loss landscape really brings here in terms of insights.

**Questions:**

- L. 135: "in order to mitigate the noisy component of forward gradients estimated by SPSA, we propose sign-m-SPSA": do you have evidence that sign-m-SPSA results in less noisy gradients?
- Section 3.3: could you please clarify what is kept in full precision? I see at least three different quantities not being quantized: i) the scales $\Delta_w$ and $\Delta_z$,  ii) the loss for each perturbed weights and therefore its difference, iii) the averaged gradient (Eq. 7). Most importantly, do you confirm that you need to accumulate gradients across each direction ($i=1 \cdots m$) in higher precision (32 bits I guess?) and then quantize-dequantize it using the scale of the perturbation $\Delta_z$? Your pseudo-algorithm only treats the case $m=1$ so it remains unclear how this all work when $m>1$ and you need to average gradients. **Could you please write a new pseudo-algorithm**, alike Alg.1, **in the case $m>1$**, highlighting with **two different color codes** the quantized (int8 and int16) and full precision (fp32) quantities?
- L.193-198 (momentum-guided sampling): I think that incorporating momentum into your approach is crucial. However, I don't understand neither how it works. What do you mean by "as training progresses, a history of the momentum $z$ is incorporated to guide the new sampling process"?
- L.199 (sharpness-aware perturbation): you mean an "extra step of **directional** gradient ascent"?
- L.204-210 (sparse updates): which sparsity scheme did you employ? top-k magnitude based scheme may be quite costly, if it boils down to ranking all the weights by their magnitude.
- L.211 (kernel-wise normalization): in this case we agree that $\hat{g}$ needs to be stored in full precision? Also, the computation of the norms of $z$ and $w$ is computationally expensive ($O(d)$, where $d$ denotes the dimension of $w$ or $z$), as expensive as it would be to dynamically recompute $\Delta_z$ and $\Delta_w$, which you avoided by statically quantizing them. Don't you lose the advantage here of using static scales if in any case you need to perform this $O(d)$ operations?
- L.216 (few-shot learning): "a few labeled samples are available", but how many? Could you please clarify the experimental setup?
- L.222: "16w8a" means 16 bits for weights and 8 for activations, correct? I may not take this for granted and clearly define this notation.
- Table 2: I would rather compute the **relative** accuracy degradation (acc_BP - acc_qFF / acc_BP) rather than the **absolute** accuracy degradation.
- Table 2: the accuracy degradation when employing FF in the FT setting compared to BP in the same setting is quite severe (11.08% gap), although you are using a relatively small architecture (ResNet12) with relatively small input dimensionality (32x32). Why is this the case?
- Table 2: **there are not any error bars**, which makes it hard to make any sense of a $\sim 0.2/0.5$ difference between two experiments.
- Could you please define precisely what you mean by "zero shot" (I assume no training at all?), "linear probing" (I assume only the last linear layer is learned?) and "full fine-tuning" (all parameters are learned)?
- L. 238 (audio benchmark): could you please detail the few-shot setup for this task, and the tasks themselves? It is important for people not familiar with this literature.
- L. 252: I really did not understand what "cross-domain adaptation" really is about. Could you please explain better what it is?
- L.256: what is "visual-prompt tuning with deep prompts"?
- Fig. 2: except for large discrepancies between bars, it is difficult to draw any conclusion from this figure **for lack of error bars**. Could you please add them?
- Which conclusions / insights do you really gain from plotting the 2D contours of the loss landscape in the different settings?

**Limitations:**

See weaknesses and questions above.

If I had detailed description of each of the experimental setups tackled, a precise knowledge of what exactly is kept in full precision, how some of the algorithmic enhancements really work and error bars on all figures, I would be prone to increasing my score. I really want to encourage the authors to do so because I do believe that the core of this work is of interest.

---

> ### Author Rebuttal · Authors · 2024-08-06
>
> We would like to express our gratitude to the reviewers for their careful review of our paper, their interest in the core idea of our work, and valuable suggestions. The comments regarding the detailed experimental setups, precision used in the algorithm, explanations of enhancement mechanisms and notations are well taken, and the manuscript will be revised accordingly. Given the large number of experiments across multiple benchmark datasets, backbones, finetuning methods, and different modalities, we need additional time to report the error bar metrics. However, we will include error bars across as many settings as possible in the final paper. Our responses to the questions are given as follows.
>
> **Question 1 (L. 135):**
>
> The design of sign-m-SPSA addresses gradient noise in two key ways.
>
> __1.  Sign-based optimizers:__ Sign-based optimizers, such as sign-SGD (Bernstein et al. [2018]), have shown good practical performance for gradient compression. In our case, the magnitude of loss changes due to perturbed weights can be quite noisy. Therefore, incorporating the sign(.) operation in m-SPSA largely reduces the noise and enhances training stability.
>
> __2. Quantization-friendly:__ Sign-m-SPSA is designed to be compatible with quantization. It constrains the range of gradient values to be the same as perturbation z for static quantization. This maintains consistency in gradient estimation within the quantized space.
>
> **Question 2 (precision):**
>
> We have updated the pseudo-algorithm with gradient averaging, and highlighted different precisions with color codes. Specifically,
>
> __1. Quantization scaling factors:__ The scaling factors $\Delta_w$ and $\Delta_z$ are floating-point values, indicating the minimum representation power of the quantization space. In fixed-point engines, these scaling values are approximated using a multiplication ($\times m$) and a right shift ($\gg k$) operations in the re-quantization stage through a post-processing hardware block, as detailed in Appendix A. For instance, $\Delta_z \approx \frac{m}{2^k} $.
>
> __2. Loss calculation:__ The loss can be computed either in floating-point precision or in quantized space, depending on the implementation and hardware support. Only the sign of the loss difference is necessary for gradient computation.
>
> __3. Gradient accumulation:__ When m>1, higher precision is used to accumulate gradients, followed by a re-quantization step. For example, if $m = 4$, we require at least 2 additional bits to store the intermediate values. Techniques such as right-shifting the gradients ($g \gg 2$) can be employed to manage memory usage while still allowing accurate accumulation. There is a trade-off between memory efficiency and accuracy.
>
> **Question 3 - 6 (enhancement techniques):**
>
> We would like to address various enhancement techniques in the common rebuttal section.
>
> **Question 7 (few-shot learning):**
>
> We use a 5-way 5-shot setting (5 labeled samples) for vision tasks and a 5-way-1-shot setting (1 labeled sample) for audio tasks.
>
> **Question 8 (16w8a):**
>
> Yes, “16w8a” denotes 16-bit weights and 8-bit activations. We will clarify the term.
>
> **Question 9 (relative metric):**
>
> Thanks for the valuable suggestion. We agree that relative accuracy degradation is a more informative metric for assessing the impact of the change, while the absolute values intuitively highlight the performance gap.
>
> **Question 10 (accuracy degradation):**
>
> The accuracy gap between BP and FF can vary based on factors such as backbone architecture, dataset and task difficulty. For instance, with the CIFAR-100 dataset, which only contains low resolution images (32x32), FF faces some challenge. Using a stronger backbone such as ViT, can help bridge this accuracy gap. This indicates that while FF may show more degradation with smaller architectures and low-resolution inputs, performance improvements can be achieved with more advanced models.
>
> **Question 11 and 16 (error bars):**
>
> It is generally expected that BP outperforms FF in terms of accuracy in most tasks. However, our goal is to narrow this gap while leveraging the memory benefits of FF. We acknowledge the importance of including error bars. We will include error bars across as many settings as possible in the updated manuscript.
>
> **Question 12 (terms of training methods):**
>
> All terms are correct. We will clearly define these terms to avoid any ambiguity.
>
> **Question 13 (L. 238):**
>
> We will provide more detailed task information in the audio benchmark section to ensure clarity for readers who may not be familiar with this literature.
>
> **Question 14 (L. 252):**
>
> 1.	"Cross-domain" refers to fine-tuning on tasks with data distribution significantly different from those of the pre-trained model. For example, a model pre-trained on the ImageNet might be adapted to perform tasks on the VWW dataset.
> 2.	"In-domain", on the other hand, refer to fine-tuning a model on tasks with data distributions more closely related to the pre-trained data, but with some variations. For example, a model pre-trained on CIFAR-100 might be adapted to handle corrupted versions of the same dataset.
>
> **Question 15 (VPT-deep):**
>
> This refers to VPT-deep, a fine-tuning method for Transformer models, as introduced by (Jia et al. [2022]).
>
> **Question 17 (2D contours):**
>
> Plotting the 2D contours of the loss landscape provides valuable insights into the training dynamics of different methods. These plots visualize how the loss evolves over training epochs and how the optimization paths differ.
>
> From the contours, we observe that both FF and BP exhibit locally smooth loss landscapes, with trajectories generally following the gradient decent direction. However, compared to BP, FF tends to take more conservative steps at the beginning of the training, resulting in slower convergence. Despite this, both methods converge to a local minimum after 100 epochs, indicating that FF, while slower, ultimately reaches a comparable solution to BP.

---

> > ### Comment · Reviewer_uc7y · 2024-08-08
> > **Answer to rebuttal**
> >
> > Dear authors,
> >
> > Thank you very much for answering my questions and adding the detailed pseudo-algorithm in the global rebuttal.
> > I am happy to increase my score to accept!

---

> > > ### Author Response · Authors · 2024-08-10
> > > **Replying to reviewer's comment**
> > >
> > > We sincerely appreciate the reviewer's recognition of the significance of our work and the increased score.
> > >
> > > Thank you!

---

### Official Review · Reviewer_vPS7 · 2024-07-11

**Soundness:** 3
**Presentation:** 2
**Contribution:** 3
**Rating:** 5
**Confidence:** 3

**Summary:**

This paper explores the feasibility of on-device training using fixed-point forward gradients. The authors propose methods including sign-m-SPSA, Momentum Guided Sampling, Sharpness-aware Perturbation, Sparse Update, and Kernel-wise Normalization to reduce memory footprint and accuracy gaps and conduct experiments across various deep learning tasks in vision and audio domains. Key contributions of this paper include formulating forward gradients in the quantized space, demonstrating the feasibility of on-device training, and visualizing the neural loss landscape during training. The study shows that training with fixed-point forward gradients might be a practical approach for model customization on edge devices.

**Strengths:**

++ This paper proposes an improved method for forward gradients, called Quantized Zeroth-order Forward Gradient (QZO-FF), which enables forward gradients training using quantization.

++ QZO-FF is quantized and does not require backpropagation, thereby reducing memory overhead and eliminating the need for processors to have training capabilities. However, I doubt this because even though forward gradients do not require backpropagation, they still need to update weights and possibly save momentum and they need to perform additional quantization for $z$ in QZO-FF. Therefore, we may need some hardware adaption to assist feed-forward training.

++ The experiments across various benchmarks show that there is only a slight degradation in accuracy while the memory cost is reduced.

**Weaknesses:**

1. Some results are missing in the experiment. For example, (1) the memory cost of (BP, LP, fp16) is not measured. I think the memory cost of LP is important because it seems that the reduction of memory cost mainly comes from LP instead of FF and Quant in Figure 3 and Figure 4, and I think the claim that "this number is further reduced to only 0.28MB" and "the saving increases to 8.1× when sparse update and fixed-point are enabled" in Appendix B is totally misleading and unfair. (2) The accuracy of (BP, LP, quant) is not measured so there is no baseline for (FF, LP, Quant). (3) The accuracy of (FF, FT, Quant) and (BP, FT, Quant) is not measured. (BP, FT, Quant) should be some BP fixed-point training methods like Quantization-Aware Scaling (QAS) mentioned in related work.

2. Lack of ablation studies. The effects of techniques proposed in Section 3.4 are not well-studied. (1) There is no ablation study for Section 4.1. (2) The effect of sharpness-aware and kernel-wise normalization is not measured separately in Section 4.2. (3) I want to know __which__ of these techniques work in __what__ experiment settings. I believe that, as a new algorithm with many enhancement techniques, the authors should inform the readers about which parts of the algorithm are useful under which circumstances.

3. The model size (100K - 80M) is somewhat small compared to the concept of "pretrained models". How does the proposed method perform for larger models and how does the model size affect the effectiveness of the method?

**Questions:**

1. Although one can understand the meaning of these symbols after a careful reading, the notation in equation (6) is somewhat confusing because $1_q$ and $\epsilon_q$ have the same subscript but different scaling factors. I think it would be better to add a notation related to the scaling factor above them.

2. typo in line 271: extenteded

**Limitations:**

The limitations are discussed well by the authors.

---

> ### Author Rebuttal · Authors · 2024-08-06
>
> We would like to express our gratitude to the reviewers for their careful review of our paper, their interest in the core idea of our work, and valuable suggestions. The comments regarding more detailed explanations of techniques, experimental setups and comparisons are well taken, and the manuscript will be revised accordingly. Our responses to the questions are given as follows.
>
> **Question 1 and 2 (notation in equation 6, typo in line 271):**
>
> Thank you for pointing this out.
>
> The subscript $q$ refers to the quantized value. In this context, $1_q$ and $\epsilon_q$ each have their own scaling factors ($\Delta_z$ and $\Delta_w$) and bit-widths.
>
> We will correct the typo, and revise the notation to clearly distinguish these elements.
>
> **Weakness 1 (some missing results):**
>
> __1.	The memory cost of (BP, LP, fp16):__
>
> Thank you for your feedback. We understand the importance of accurately representing the memory costs associated with different training methods. The extent of memory saving with FF depends on the number of layers being fine-tuned, and their positions within the network. When applied to methods such as Full Fine-tuning (FT), LoRA and other Parameter-Efficient Fine-tuning (PEFT) approaches, FF shows significant memory reduction because it eliminates the need to store intermediate activations. This is evident when comparing the memory usage of (FF, FT) vs (BP, FT). In contrast, for Linear Probing (LP), where only the last layers are updated, the memory savings are less. Typically, finetuning with LP results in lower accuracy compared to FT or LoRA.
>
> We will revise our statements and ensure that the claims are fair and well-supported.
>
> __2.	The accuracy of (BP, LP, Quant) and (BP, FT, Quant):__
>
> We use (BP, LP, fp16) as a higher accuracy baseline to compare with (FF, LP, Quant). With techniques such as Quantization-Aware Scaling (QAS) reported in the literature, we expect (BP, LP, Quant) to be close of (BP, LP, fp16) in terms of accuracy. Similarly, we expect that (BP, FT, Quant) to perform comparably to (BP, FT, fp16). Currently, support for Quant formats with BP is limited on most fixed-point engines.
>
> **Weakness 2 (Ablation studies):**
>
> __1.	Ablation study for Section 4.1__
>
> We conducted ablation studies in Section 4.2 to evaluate the effectiveness of quantized FF, the impact of bit-width variations, and different perturbation sampling strategies. This section provides a more controlled experimental setting for these analyses.
>
> In Section 4.1, we focused on evaluation few-shot learning across a diverse set of benchmark datasets, tasks, backbones, and modalities. This broader approach aims to demonstrate the general applicability of our method across various scenarios, while the more detailed ablation studies in Section 4.2 address specific aspects of our approach.
>
> __2.	The measure of sharpness-aware and kernel-wise normalization:__
>
> In our cross-domain adaptation setup, we observed a consistent, but only a modest increase in accuracy with the combination of sharpness-aware and kernel-wise normalization.
>
> __3.	Various techniques and experiment settings:__
>
> Thank you for raising this important point. We would like to address this comment in the common rebuttal section, providing a detailed explanation of how each technique performs. We will include a discussion of each technique and its effectiveness across different experimental settings in the updated manuscript, to guide readers on the optimal use of these enhancements.
>
> **Weakness 3 (QZO-FF for larger models):**
>
> Thank you for the suggestions. We will address this comment in the common rebuttal section.

---

> ### Comment · Reviewer_vPS7 · 2024-08-12
>
> Thanks for the authors' response! Some of my concerns haven't been addressed so I keep my score.
>
> **The memory cost of (BP, LP, fp16)**
>
> Will the authors provide the memory cost of (BP, LP, fp16)? How will the author modify their statements?
>
> **Ablation study for Section 4.1**
>
> Why did the author only provide an ablation study in Section 4.2 and not include one in Section 4.1?
>
> **QZO-FF for larger models**
>
> The authors haven't conducted experiments on models larger than 80M so I think the concept of "pre-trained models" in the abstract (line 1) and the introduction (line 20) is overclaimed.

---

> > ### Author Response · Authors · 2024-08-12
> > **Replying to reviewer's comment**
> >
> > We sincerely appreciate the reviewer’s comment on the fair comparison of memory cost of BP and ZO-FF among various training methods. To address this, we would like to make some clarifications.
> >
> > __1. Memory cost of BP and ZO-FF__
> >
> > * For a network with $N$ layers, BP with FT consumes $O(N)$ memory while ZO-FF always uses $O(1)$ memory. With an increased number of $N$, ZO-FF benefits more on memory.
> > * In the case of LP, where only the last layers are updated, it is expected that the difference of memory usage between BP and ZO-FF will be very small. For example, in our ViT tiny network for vision tasks, the total memory usage and scratch memory usage of (BP, LP, fp16) is $11.81MB$ and $0.45MB$, respectively. This number is very close to that of (FF, LP).
> > * We will provide an extra bar in Figure 3 and 4 for the case of (BP, LP, fp16), and make it clear that the comparison of memory is among the same training method. We do not compare memory usage of (BP, FT) vs (FF, LP), since they are updating different number of layers in network.
> >
> > __Ablation studies:__
> >
> > We choose Section 4.2 to study the impact of various factors on ZO-FF. This section has a more controlled experimental setting (ViT tiny backbone, and VWW dataset) between BP and ZO-FF. In our ablation studies, we keep all the settings the same and vary only one factor at a time. In Section 4.1, we would like to focus on the few-shot learning benchmark results, where broader experimental settings are used, including various backbone network architectures, datasets, and vision/audio modalities.
> >
> > __ZO-FF for larger models:__
> >
> > We apologize for any confusion regarding "pre-trained models". In our work, the pre-trained model refers to models that are pre-deployed on the device, for instance, an object detection model. These models may need additional adaptation or personalization over time.
> >
> > The primary focus of our work is to enable such model fine-tuning on existing edge devices with fixed-point engines (e.g., NPUs, DSPs, MCUs). These hardware are primary designed for inference, therefore, typically have very limited memory and lack support for BP. Implementing BP on such hardware requires substantial engineering efforts. Under this context, ZO-FF directly leverages fixed-point forward calls for gradient estimation, with the same memory cost as inference, facilitating model fine-tuning without hardware adaptation. With all the benefits of ZO-FF, we believe that it is an attractive point along the compute-memory trade-off, and especially suitable for on-device fine-tuning use cases when the limited memory is the main stumbling block.
> >
> > Thank you!

---

> > > ### Comment · Reviewer_vPS7 · 2024-08-13
> > >
> > > Thanks for the response!
> > >
> > > Most of my concerns have been addressed except that I'm not convinced by the answer to "why there is no ablation study in Section 4.1". So I keep my score as 5 but vote for acceptance.

---

> > > > ### Author Response · Authors · 2024-08-13
> > > > **Replying to reviewer's comment**
> > > >
> > > > We appreciate the reviewer’s valuable comments and suggestions, and we are glad that most concerns have been addressed.
> > > >
> > > > In Section 4.1, we conducted a total number of 115 experiments across two modalities. Due to the number of experiments and the training time required for each, performing ablation studies for all cases is not practical within the given timeline. However, if time allows, we can select one use case and conduct ablation studies similar to those in Section 4.2.
> > > >
> > > > We welcome any suggestions on which use case would be of particular interest to the research community.
> > > >
> > > > Thank you!

---

### Official Review · Reviewer_oWX1 · 2024-07-11

**Soundness:** 2
**Presentation:** 3
**Contribution:** 2
**Rating:** 5
**Confidence:** 3

**Summary:**

The authors investigate fixed-point forward gradients for quantized training. They conduct experiments across various deep learning tasks in vision and audio to assess if this method yields competitive models while conserving memory and computational resources.
They introduce algorithm enhancements to reduce memory usage and accuracy gaps compared to backpropagation, using fixed-point precision for forward gradients during training or adaptation.
Their findings demonstrate the feasibility of on-device training with fixed-point forward gradients across diverse model architectures (e.g., CNN, RNN, ViT-based) and parameter sizes (100K to 80M), offering practical solutions for model adaptation on edge devices.
The authors also visualize neural loss landscapes and training trajectories, providing insights into the dynamics of training with forward gradients for efficient on-device model adaptation.

**Strengths:**

1 .They understand quantization and tried not to leave anything float

2. Experimenting with SAM and ZO is nice

3. The paper is well written

**Weaknesses:**

1. Sadly no experiments on LLMs on which most fine tuning is done today

2. Marginal novelty: generally they just added quantization to ZO-FF – is that enough?

**Questions:**

1. why you loop over w can’t you just do it vector wise?

2. Can you specify m (the number of pertubations) used for each experiment?

3. Can you calculate the memory consumption (MB/GB)  and computation complexity (in FLops) compared to QLoRA with BP.

**Limitations:**

yes

---

> ### Author Rebuttal · Authors · 2024-08-06
>
> We would like to express our gratitude to the reviewers for their careful review of our paper, their interest in the core idea of our work, and valuable suggestions. The comments regarding the motivation, novelty, impact of our work, and detailed comparisons of hardware complexity are well taken, and the manuscript will be revised accordingly. Our responses to the questions are given as follows.
>
> **Question 1 (loop over $w$):**
>
> The loop of $w$ pertains to performing updates on a per-tensor basis. Each individual tensor $w_i$ is the trainable parameters for each layer of the model. It is processed in a vectorized manner, and is associated with its own quantization scaling factor. We will clarify this notation in the algorithm description to ensure a better understanding of the process.
>
> **Question 2 (specify of $m$):**
>
> In our experiments, the number of forward-forward calls performed ($m$) for averaging gradients is set to $3$, unless otherwise specified in the ablation studies. Details on this parameter are provided in the appendix.
>
> **Question 3 (memory consumption and computation complexity):**
>
> We appreciate your recommendations and feedback. We will list the memory cost and FLOPs for all models used in backpropagation (BP) and forward-forward (FF) process in the updated manuscript. Specifically,
>
> We compare BP and FF from two perspectives:
>
> __1. Memory efficiency:__ The extent of memory saving with FF depends on the number of layers being fine-tuned, and their positions within the network. When applied to methods such as Full Fine-tuning, LoRA and other Parameter-Efficient Fine-tuning (PEFT) approaches, FF shows significant memory reduction because it eliminates the need to store intermediate activations.
>
> __2. Computation complexity:__ For a single iteration, BP performs one forward pass and one backward pass, while FF needs two forward passes. The FLOPs of a backward pass are roughly 2x of that of the forward pass (e.g., for both Convolutional and Linear layers). In our experiments, we observed a 1.5x speedup in one iteration of the training. However, the total computation (or training time) depends on the number of iterations required for the training to be converged.
>
> **Weakness 1 (QZO-FF for LLM models):**
>
> We would like to address this comment in the common rebuttal section.
>
> **Weakness 2 (novelty):**
>
> Most existing neural processors on edge devices are optimized as efficient fixed-point inference engines. We believe continuously adapting pre-trained models to local data on the edge is crucial for effective model deployment.
>
> To enable training on edge devices with constrained memory, we leverage low-bit precision techniques. While previous work by [J. Lin, 2022], introduced Quantization-Aware Scaling for BP, to mitigate accuracy loss due to quantization, BP is memory intensive, and unsupported by many existing inference engines. In contrast, ZO-FF shows a significant advantage in this context.
>
> To our knowledge, there is no prior research demonstrating the feasibility and impact of quantized ZO-FF, particularly with regard to different bit-widths and their effects on performances. Our work addresses this gap by showing the effectiveness of quantized ZO-FF through extensive experiments across various deep learning benchmarks and modalities. We believe this contribution is of substantial interest to the industry, addressing both practical challenges and opportunities in deploying models on edge devices.

---

> > ### Comment · Reviewer_oWX1 · 2024-08-11
> > **Answer to the Authors rebuttal**
> >
> > I would like to thank the authors for their detailed response. However, I still believe that a fair comparison of memory and computational complexity should include the number of steps required to achieve a certain level of accuracy. Currently, when a practitioner has a system with limited memory, they often use activation checkpointing, a technique used to manage memory consumption in large language models (LLMs) during training. Instead of storing all intermediate activations for backpropagation, only a subset of activations is saved at "checkpoints." The rest of the activations are recomputed as needed during the backward pass. This approach reduces memory usage significantly at the cost of increased computation time. If QZO-FF requires three perturbations and converges much more slowly, say five times slower to achieve same accuracy, it might not be beneficial to use it. Can you add convergence curves for BP and QZO-FF or numbers of iterations required to reach the reported accuracy? If so I'll consider raising my  score.

---

> > > ### Author Response · Authors · 2024-08-12
> > > **Replying to reviewer's comment**
> > >
> > > We sincerely appreciate the reviewer’s comment on the comprehensive comparison of QZO-FF and BP regarding memory, computational complexity, and convergence speed. We agree that these factors are essential for evaluating different gradient calculation methods. To address these aspects, we will provide training curves, along with reporting the accuracy of each method in our ablation studies, and include a time-compute-memory trade-off analysis comparing BP, BP with checkpointing and QZO-FF. Specifically,
> > >
> > > __1.	Convergence speed and empirical measurements__
> > >
> > > Our experiments on the ViT tiny network with $5.2M$ parameters shows that ZO-FF converges approximately $2$x slower than BP, for both LP and VPT-deep training methods. Unfortunately, due to the format limitation, we cannot upload any link or graphs in the current response page. However, we will include the training curves in the Appendix to illustrate the convergence of BP and QZO-FF across various settings (e.g., $m=1$, $m=3$ for gradient averaging) and different precisions (fp16, quant). These curves will show the actual convergence behavior and training time required to achieve the reported accuracy.
> > >
> > > __2.	Compute-Memory Trade-off__
> > >
> > > Analyzing the time-memory trade-off for backpropagation (BP) is complex. However, the work of [Griewank et al. 2008] gives a general rule of time-memory trade-off for BP (Rule 21). For a network with $N$ layers, and a time-memory tradeoff hyperparameter $c = O(1)$, there exists a BP algorithm that runs in $O(cN)$ time and consumes memory proportional to $O(N^{1/c})$.
> > >
> > > * In the case of $c=1$ (storing everything during the forward path), BP consumes $O(N)$ compute and $O(N)$ memory. Given the FLOPs of a backward pass are roughly $2$x of that of the forward pass, ZO-FF (with $m=3$) consumes $O(2N)$ compute and $O(1)$ memory, so it uses more compute but substantially less memory.
> > >
> > > * Gradient checkpointing [Chen et al. 2016] reduces memory cost of BP by recomputing some activations. In their experiments, choosing $c=2$ achieves $O(\sqrt{N})$ memory at $O(2N)$ computation. In comparison, ZO-FF is more compute-efficient at the same memory cost.
> > >
> > > We want to emphasize that the memory cost of gradient checkpointing is bounded by ZO-FF and BP, and FF is more compute-memory efficient at the same memory cost as gradient checkpointing along the pareto curve.
> > >
> > > As an example, the memory cost of training a ViT tiny network using different gradient estimation methods is illustrated as following (input size 224x224x3, batch size 1, and FP16 data type. QZO-FF is using16w8a). For BP with gradient checkpointing, we assume half of the activations are stored during forward time.
> > >
> > > | **methods** | **weights** (MB)     | **activations (peak + stored)** (MB)          | **weight gradients** (MB)     |
> > > |---------------------------|-----------------------|----------------------------|-----------------------|
> > > | **BP**                    | ██████████ 10.54       | ▓▓▓▓▓▓▓▓▓▓▓▓▓ 19.56        | ▒▒▒▒▒▒▒▒▒▒ 10.54       |
> > > | **BP_grad_checkpointing** | ██████████ 10.54       | ▓▓▓▓▓▓▓ 9.92            | ▒▒▒▒▒▒▒▒▒▒ 10.54       |
> > > | **ZO_FF**                 | ██████████ 10.54       | ▓ 0.14                     | ▒▒▒▒▒▒▒▒▒▒ 10.54       |
> > > | **QZO_FF**                | ██████████ 10.54       |  0.07                     | ▒▒▒▒▒ 5.27           |
> > >
> > >
> > > __Additional comments:__
> > >
> > > Beyond memory savings, QZO-FF provides practical benefits in several areas: 1) it enables training on existing edge devices with fixed-point engines (e.g., NPUs, DSPs, MCUs), which typically have very limited or no support for BP, because these hardware are primarily designed for inference. Implementing BP on such hardware requires substantial engineering efforts. Additionally, the required memory to run training is critical, determining the feasibility of enabling such feature on device. QZO-FF directly leverages fixed-point forward calls for gradient estimation, with the same memory cost as inference, facilitating model fine-tuning without hardware adaptation. This is a primary focus of our work. 2)  QZO-FF is also suitable for training with non-differentiable objectives (e.g., maximizing accuracy, F1-score), where BP cannot be directly applied.
> > >
> > > In summary, we do not expect QZO-FF to outperform BP in convergence speed or as a replacement of BP, and we will include the above analysis to guide readers on the optimal choose of these techniques. With all the benefits of QZO-FF, we believe that it is an attractive point along the compute-memory pareto curve and especially suitable for on-device fine-tuning use cases when the limited memory is the main stumbling block.
> > >
> > > Thank you!

---

### Official Review · Reviewer_wcC5 · 2024-07-18

**Soundness:** 3
**Presentation:** 3
**Contribution:** 3
**Rating:** 5
**Confidence:** 4

**Summary:**

The paper proposes a quantization approach for fine-tuning pretrained data to new local data on resource-constrained devices. In particular, the weights perturbation, gradients estimation, and weights updates are quantized to either 8-bit or 16-bit. This quantization approach is combined with Momentum Guided Sampling, Sharpness-aware Perturbation, Sparse Update, and Kernel-wise Normalization to enhance fine-tuning performance. The proposed approaches are evaluated on various AI benchmarks. The results of this study indicate that quantized forward gradients are a good candidate for a fine-tuning approach that can be deployed on edge devices.

**Strengths:**

1- The paper is well-written and well-organized.

2- The quantized approach is evaluated on a variety of tasks that show the generalizability of the new approach.

3- The Sign-m-SPSA-SGD approach is interesting and novel.

**Weaknesses:**

1- The author are recommended to discuss the accuracy degradation of quantized forward gradients compared to the backpropagation algorithm. In some cases, the accuracy degradation is high (more than 5%). A comparison of performance versus hardware complexity (FLOPs or another metric) is recommended, as seen in [1].

2- Evaluating the efficacy of quantized forward gradients on fine-tuning LLM models such as LLaMA-3 is recommended.

[1] Carmichael, Zachariah, et al. "Performance-efficiency trade-off of low-precision numerical formats in deep neural networks." Proceedings of the conference for next generation arithmetic 2019. 2019.

**Questions:**

Why is the random perturbation vector z sampled from a normal distribution with zero mean and standard deviation? Is it possible to sample from a log-normal distribution since activation gradients are shown to be distributed near log-normal [1]?

[1] Chmiel, Brian, et al. "Neural gradients are near-lognormal: improved quantized and sparse training." arXiv preprint arXiv:2006.08173 (2020).

**Limitations:**

The author addresses the limitations of this study by mentioning the initialization requirements for forward gradients and 16-bit weight quantization.

1- It would be suggested to discuss various initialization approaches that might be used instead of a pretrained network.
2- It would be suggested to discuss other numerical formats, such as 8-bit floating point or Posit, to solve the 16-bit weight quantization problem.

---

> ### Author Rebuttal · Authors · 2024-08-06
>
> We would like to express our gratitude to the reviewers for their careful review of our paper, their interest in the core idea of our work, and valuable suggestions. The comments regarding accuracy discussions and comparisons of hardware complexity are well taken, and the manuscript will be revised accordingly. Our responses to the questions and identified weaknesses are given as follows.
>
> **Question 1 (sampling of $z$):**
>
> The choice of sampling $z$ from a normal distribution with zero mean and unit variance is supported by the literature [Baydin et al. [2022], Section 3.2], which proved that the forward gradient is an unbiased estimator of the true gradient, when the scalar components of perturbation are independent, and follow a zero mean, unit variance distribution.
>
> In our case, Sign-m-SPSA is also designed to be compatible with quantization. It constrains the range of gradient values to be the same as perturbation $z$ for static quantization. This maintains consistency in gradient estimation within the quantized space.
>
> In addition to using a normal distribution, we explored using a binomial distribution from a quantization-friendly perspective. Our experiment, as shown in Figure 2(a), indicates that the binomial distribution is also effective. Sampling from other distributions (e.g., log normal distribution) is another interesting direction. We tested this variation on our cross-domain adaptation setup with fp16 precision, and the results were promising. The quantization impact of such distribution needs to be further investigated.
>
> **Weakness 1 (accuracy and hardware complexity):**
>
> We appreciate your recommendations and feedback. We will include a more detailed accuracy analysis, and list the memory cost and FLOPs of all the models used for backpropagation (BP) and forward-forward (FF) in the updated manuscript. Specifically,
>
> __1. Accuracy Discussion:__
>
> Since FF solely utilizes directional derivatives for gradient estimation, it is expected that BP generally outperforms FF in terms of accuracy in most tasks. The accuracy gap between BP and FF can vary based on factors such as the backbone architecture, dataset, and task difficulty. We observed that accuracy gap tends to increase on more challenging tasks. However, using a stronger backbone such as ViT, can help bridge this gap. This indicates that while FF may show more degradation with smaller architectures and low-resolution inputs, performance improvements can be achieved with more advanced models.
>
> __2. Hardware Complexity:__
>
> We compare BP and FF from two perspectives:
>
> * __Memory efficiency__: BP needs memory to store model parameters, all the intermediate activations and gradients, whereas FF avoids storing intermediate activations, the size of which could be considerably large in many models.
>
> * __Computation complexity__: For a single iteration, BP performs one forward pass and one backward pass, while FF needs two forward passes. The FLOPs of a backward pass are roughly 2x of that of a forward pass (e.g., for both Convolutional and Linear layers). In our experiments, we observed a 1.5x speedup in one iteration of the training. However, the total computation (or training time) depends on the number of iterations required for the training to be converged.
>
> **Weakness 2 (QZO-FF for LLM models):**
>
> We would like to address this comment in the common rebuttal section.
>
> **Limitation discussion 1 (initialization):**
>
> Our work primarily focuses on model fine-tuning on edge devices with fixed-point engines, where we assume that the pre-trained network provides a good initialization. To facilitate reproducibility, we use widely available open-source pre-trained networks as backbones. In our cross-domain adaptation experiments, we initialize the decoder layers randomly to ensure a consistent and straightforward experimental setup. We have added an accuracy report with mean and standard deviations across 5 runs to our ablation studies in Section 4.2.
>
> **Limitation discussion 2 (numerical formats):**
>
> Thank you for the valuable suggestions. Based on our extensive investigations, 16-bit weight quantization is crucial for accurately capturing perturbations and accumulating weight changes in FF. However, we recognize the potential benefits of exploring alternative numerical formats such as 8-bit floating point, or ultra-low bit formats, in forward gradient learning.
>
> In future work, we plan to explore these numerical formats to address challenges associated with 16-bit weight quantization. Currently, support for such formats is limited on most fixed-point engines.
>
> We will include additional discussions in the updated manuscript.

---

### Author Rebuttal · Authors · 2024-08-06

We would like to express our gratitude to the reviewers for their careful review of our paper, and their interest in the core idea of our work. We appreciate all the feedback, valuable suggestions and recommendations. The comments regarding notations, technical discussions, experimental clarifications are well taken, and the manuscript will be revised accordingly. Our responses to each reviewer’s questions are submitted separately. Additionally, we would like to address a few comments as follows.

**Suggestion of evaluating QZO-FF on LLM models:**

Thank you for the valuable suggestion. Our paper currently focusses on the feasibility and the effectiveness of quantized ZO optimization for fine-tuning smaller models on edge devices with fixed-point engines. We have deployed our method on memory-constrained edge devices, and brought the training capability to fixed-point engines.

While our current work centers on these smaller models (ConvNets, and Transformer models), we recognize the potential application of our approach to larger LLM models such as LLaMA-3. Techniques Like LoRA and other Parameter-Efficient Fine-Tuning (PEFT) methods could be combined with our quantized ZO approach for LLMs. Recent literature, such as the MeZO work, indicates that ZO training can be effective across various LLM tasks. However, the impact of quantized ZO with low precision for LLMs remains an open question. We plan to extend our research to evaluate the performances of our quantized ZO approach on LLM models and various benchmark tasks in future work.

**Clarification of various enhancement techniques:**

These enhancement techniques are optional, and often involve trade-offs between memory, computation and accuracy, depending on the hardware memory budget. We will provide a more detailed discussion of each technique and its effectiveness across different experimental settings in the revised manuscript, to guide readers on the optimal use of these enhancements. Specifically,

__1.	Momentum-guided sampling:__

This enhancement introduces memory overhead to increase the accuracy. Instead of sampling solely from a zero-centered Gaussian distribution, perturbations are computed from a combination of a momentum-centered and a zero-centered Gaussian distribution. Mathematically, $z_1 \sim \mathbb{N}(0, \mathbb{I}_n* \sqrt{\alpha})$, $z_2 \sim \mathbb{N}(z_t, \mathbb{I}_n* \sqrt{1-\alpha})$, and
$z _{t+1} = \beta*z_1 + (1-\beta)*z_2$. Here, $\beta$ is a smoothing parameter; $\alpha$ and $\beta$ can be adaptively adjusted during training. While $\beta=1$ corresponds to the baseline version without momentum-guided sampling, this approach enhances performance by improving perturbation quality through sampling history.

__2.	Sharpness-aware perturbation:__

This technique improves the forward gradient direction by performing an addition step of directional gradient ascent, targeting regions where the loss curve is steeper. We observed that incorporating sharpness-aware optimization enhances the overall performance of QZO-FF across various experiments.

__3.	Sparse updates:__

We base our sparsity update approach on recent work by (Chen et al. [2024]), which utilizes a zero-order sparsity method. We also experimented with a random sparsity scheme, which proved effective as well. Due to the intrinsic properties of ZO method, reducing the number of parameters updated per iteration generally leads to improved accuracy.

__4.	kernel-wise normalization:__

The primary motivation to incorporate static quantization is to ensure compatibility with hardware support. Many existing fixed-point neural processors on edge devices only support static graph quantization, where weights and activations are quantized prior to compilation. Dynamic quantization, which involves recalculating quantization parameters during runtime, is computationally expensive, and often not supported efficiently.

For kernel-wise normalization, obtaining the norm of weights involves a trade-off between computation and accuracy. However, efficient implementations using GEMM and SQRT operations can minimize the overhead on hardware.

**Figures:**

We have updated our pseudo-algorithm to include gradient averaging when $m>1$, and highlighted different precisions with color codes. Please noted that the scaling factor is approximated through a multiplication and a right shift operation on fixed-point processors (Appendix A). An accuracy report with mean and standard deviations has been added to our ablation studies in Section 4.2. Please refer to attached PDF for the updates.

---

### Comment · Area_Chair_tnQH · 2024-08-08
**Start reviewer-author discussions right now**

Dear reviewers,

Authors submitted rebuttals, which should be visible to you now.

Please read the rebuttals carefully and start discussions with the authors now.

The reviewer-author discussion period will end on August 13, 2024. Since authors usually need time to prepare for their responses, your quickest response would be very appreciated.

In case you had requested additional experiments / analysis and the authors provided in the rebuttal, please pay extra attention to the results.

Thank you,
Your AC

---

> ### Comment · Area_Chair_tnQH · 2024-08-13
> **Reviewer-author discussions will end in about 30 hours**
>
> Dear reviewers,
>
> This is the final reminder for the reviewer-author discussions.
> It will end on August 13 11:59 AoE, and then we will start AC-reviewer discussions.
>
> If you have already concluded the discussions with the authors, thank you so much!
>
> If you have not responded to the author rebuttal yet, please do so immediately. We have been waiting for your response.
>
> In case you missed it, the general author rebuttal includes a PDF file.
>
> Best,
> Your AC

---

### Decision · Program_Chairs · 2024-09-25

**Decision:**

Accept (poster)

**Comment:**

We received author rebuttals, all reviewers except Reviewer wcC5 acknowledged the rebuttals and left follow-up comments.
They also participated in AC-reviewer discussions and shared their final thoughts.

In summary, a key concern from the reviewers is limited novelty e.g.("added quantization to ZO-FF" by Reviewer oWX1), which I partially agree to. However, I found that their claimed novelty is of the proposed method. While novelty of the method is often discussed, I believe that novelty of work is more important. This work not only introduces a new proposed algorithm but also performs comprehensive experiments, exploring experimental settings, data modalities and architectures. I appreciate this effort as it should be critical to cover various datasets / modalities (vision and audio) / models in experiments to support the authors' claims.  Reviewer uc7y values the contribution as well.

Reviewers oWX1 and wcC5 suggested lack of (L)LM evaluations as a weakness of this work, but I disagree to the assessment. Since this work is focused on on-device training, training even part of (L)LMs may be technically impossible for relatively weak devices with respect to cloud instances.

Overall, this paper introduces an interesting, practical approach and demonstrates the potential of the approach through various experiments. I recommend accepting this paper for NeurIPS'24.
Last but not least, Reviewer vPS7 pointed out as a weakness that model size in this work is relatively small (100K - 80M) for "pretrained models", which I find fair and not well addressed by the general author rebuttal. The weakness point itself seems not critical enough to reject this paper, but I think it is an important discussio for future work n as the model size has been increasing.